# Generalizing Linear Autoencoder Recommenders with Decoupled Expected Quadratic Loss

**Ruixin Guo** [1*], **Xinyu Li** [1*], **Hao Zhou** [1], **Yang Zhou** [2], **Ruoming Jin** [1†]
[1]Kent State University, [2]Auburn University
{rguo5, xli74, hzhou6, rjin1}@kent.edu, yangzhou@auburn.edu

## Abstract

Linear autoencoders (LAEs) have gained increasing popularity in recommender systems due to their simplicity and strong empirical performance. Most LAE models, including the Emphasized Denoising Linear Autoencoder (EDLAE) introduced by (Steck, 2020), use quadratic loss during training. However, the original EDLAE only provides closed-form solutions for the hyperparameter choice $b = 0$, which limits its capacity. In this work, we generalize EDLAE objective into a Decoupled Expected Quadratic Loss (DEQL). We show that DEQL simplifies the process of deriving EDLAE solutions and reveals solutions in a broader hyperparameter range $b > 0$, which were not derived in Steck's original paper. Additionally, we propose an efficient algorithm based on Miller's matrix inverse theorem to ensure the computational tractability for the $b > 0$ case. Empirical results on benchmark datasets show that the $b > 0$ solutions provided by DEQL outperform the $b = 0$ EDLAE baseline, demonstrating that DEQL expands the solution space and enables the discovery of models with better testing performance.

## 1 Introduction

In recent years, deep learning has emerged as the dominant paradigm in recommendation systems, leading to increasingly complex models. However, a growing body of empirical evidence reveals a surprising trend: simple linear models often perform comparably to, or even outperform, their deep learning counterparts (Dacrema et al., 2019). In particular, linear autoencoder-based methods such as SLIM (Ning & Karypis, 2011), EASE (Steck, 2019), EDLAE (Steck, 2020), ELSA(Vančura et al., 2022), RLAE and RDLAE (Moon et al., 2023) have demonstrated strong performance, frequently surpassing more sophisticated deep models (Dacrema et al., 2021), particularly in sparse data reconstruction tasks (Monteil et al., 2024; Volkovs et al., 2017).

These methods typically aim to learn an item-to-item similarity matrix $W \in \mathbb{R}^{n \times n}$ to reconstruct the (sparse) binary user-item interaction matrix $R \in \{0, 1\}^{m \times n}$ with $m$ users and $n$ items. The reconstruction is given by $RW$, where $W$ can be viewed as a linear autoencoder (LAE) acting as both encoder and decoder. One representative example is EASE (Steck, 2019), which learns $W$ by minimizing the following objective function:

$$f(W) = \|R - RW\|_F^2 + \lambda \|W\|_F^2 \quad \text{s.t. diag}(W) = 0 \tag{1}$$

where $\|R - RW\|_F^2$ denotes the reconstruction error and $\lambda \|W\|_F^2$ is the $L_2$ regularizer. The zero-diagonal constraint $\text{diag}(W) = 0$ is imposed to prevent $W$ from collapsing to the identity mapping, in which case each item would trivially reconstruct itself. The minimizer of (1), as well as those of most LAE-based models, admits a closed-form solution. This enables greater computational efficiency compared to iterative optimization methods (e.g., gradient descent) (Bertinetto et al., 2019), while also facilitating reproducibility (Appendix G.4)..

Despite their empirical success, these models treat observed interactions (the 1s in $R$) as fixed during training, without explicitly accounting for the statistical nature of the evaluation process. In

---

*Equal contribution. [†] Corresponding author.

practice, the test set is typically constructed by *holding out* a fraction of interactions (Section 7.4.2, (Aggarwal, 2016)), so that the model is evaluated on randomly masked interactions. This discrepancy motivates adopting a statistical perspective to redesign the training objective: interactions can be modeled as random variables sampled from an underlying distribution, and the objective can be formulated in expectation, thereby better aligning the training procedure with the testing scenario.

EDLAE (Steck, 2020) provides an important precursor in this direction. By introducing *dropout* and an *emphasis weighting* scheme, EDLAE effectively reshapes the loss to penalize the reconstruction of masked interactions more heavily, thereby mitigating overfitting toward the identity mapping, see (3). Importantly, we observe that (3) can be reformulated in expectation form:

$$f(W) = \mathbb{E}_\Delta \left[ \|A \odot (R - (\Delta \odot R)W)\|_F^2 \right] \tag{2}$$

where $A \in \{a, b\}^{m \times n}$ denotes an emphasis matrix with parameters $a \geq b \geq 0$, which assigns greater weight to the reconstruction of dropped items. This statistically formulated objective closely resembles the *mean squared error (MSE)* used in evaluation (Appendix G.2). Such training-testing alignment has been empirically shown to be effective in achieving strong performance.

However, the theoretical foundation underlying the construction of this objective remains largely underexplored. In particular, Steck (2020) provides a closed-form solution for the $W$ minimizing (2) only in the special case $b = 0$. The behavior of the solution in the broader hyperparameter range $b > 0$ remains unexplored, and it is unclear whether improved models may arise in this region. Motivated by this gap, this paper studies the existence and uniqueness of closed-form solutions over the full hyperparameter range $b \geq 0$, develops efficient methods for computing them, and conducts extensive experiments to evaluate their empirical performance.

First, we generalize the EDLAE objective (2) into a **Decoupled Expected Quadratic Loss (DEQL)** and derive its closed-form minimizer, which subsumes EDLAE as a special case. This generalization not only simplifies the derivation of EDLAE solutions, but reveals several new theoretical insights: for $b = 0$, the closed-form minimizer is not unique – solutions share identical off-diagonal entries while allowing arbitrary diagonal; for $b > 0$, a unique closed-form solution always exists, including the previously unexplored region $b > a$ (Section 3).

Next, we find that the direct computation of solutions for $b > 0$ have an $O(n^4)$ time complexity, which is prohibitively expensive for large-scale recommendation tasks. To overcome this challenge, we develop an efficient algorithm based on Miller's matrix inverse theorem (Miller, 1981), reducing the complexity to $O(n^3)$. This makes computing solutions in the $b > 0$ region practical (Section 4).

Finally, we evaluate solutions derived from DEQL on real-world benchmark datasets. Our experiments demonstrate that solutions with $b > 0$, combined with $L_2$ regularization, consistently outperform the original EDLAE solutions with $b = 0$, as well as other recent LAE-based and deep learning-based recommender models. These results confirm that expanding the solution space can yield models with stronger generalization performance. Moreover, we find that the previously assumed constraint $a \geq b$ does not universally guarantee optimal performance: on certain datasets, the best-performing models lie in the range $b > a$ (Section 5).

Appendix A shows an illustration of DEQL framework. The proofs of all theorems, lemmas and propositions are in Appendix B. Related works are in Appendix E. Discussions are in Appendix G.

## 2 PRELIMINARIES

**LAE-based Recommender Systems**: In the implicit feedback setting, a dataset is typically represented as a binary user-item interaction matrix $R \in \{0, 1\}^{m \times n}$ of $m$ users and $n$ items. Each $R_{ij}$ indicates whether user $i$ has interacted with item $j$. $R_{ij} = 1$ is referred to as an observed entry, meaning that user $i$ has interacted with item $j$; $R_{ij} = 0$ is referred to as a missing entry, meaning no observed interaction. Collaborative filtering recommender systems aim to identify items that a user has not yet interacted with but is likely to interact with, and recommend them accordingly. To this end, the model predicts the missing entries in $R$ by assigning real-valued scores, where larger values indicate a higher likelihood of interaction.

LAEs are one class of collaborative filtering models. They learn a parameter matrix $W \in \mathbb{R}^{n \times n}$ and generates predictions by $\hat{R} = RW \in \mathbb{R}^{m \times n}$, so that each missing entry $R_{ij} = 0$ is assigned a predicted score $\hat{R}_{ij} \in \mathbb{R}$.

**EDLAE** (Steck, 2020): EDLAE is a state-of-the-art LAE model for collaborative filtering. Let $\Delta \in \{0,1\}^{m \times n}$ be a random matrix whose entries $\Delta_{ij}$ are i.i.d. Bernoulli random variables with $P(\Delta_{ij} = 0) = p$ and $P(\Delta_{ij} = 1) = 1 - p$ for a given $p \in (0,1)$. Let $\Delta^{(k)}$ denote a realization of $\Delta$ with index $k$, and let $\odot$ denote the Hadamard (element-wise) product, so that $\Delta^{(k)} \odot R$ applies dropout element-wise to $R$. Define the emphasis matrix $A^{(k)}$ where $A_{ij}^{(k)} = a$ if $\Delta_{ij}^{(k)} = 0$ and $A_{ij}^{(k)} = b$ if $\Delta_{ij}^{(k)} = 1$. Then, the EDLAE model is obtained by optimizing the following objective function:

$$W^* = \arg\min_{W} \lim_{N \to \infty} \frac{1}{N} \sum_{k=1}^{N} \|A^{(k)} \odot (R - (\Delta^{(k)} \odot R)W)\|_F^2 \qquad (3)$$

under the hyperparameters $a, b, p$. Since the squared Frobenius norm in (3) can be expanded into the sum of weighted quadratic loss $\sum_{i=1}^{m} \sum_{j=1}^{n} A_{ij}^{(k)^2} (R_{ij} - (\Delta_{i*}^{(k)} \odot R_{i*})W_{*j})^2$, if $R_{ij}$ is dropped, its reconstruction loss $(R_{ij} - (\Delta_{i*}^{(k)} \odot R_{i*})W_{*j})^2$ is weighted by $a^2$; otherwise, it is weighted by $b^2$.

The original EDLAE paper (Steck, 2020) suggests choosing $a, b$ to satisfy $a \geq b \geq 0$, and provides a closed-form solution to (3) for the case $b = 0$ and under the zero-diagonal constraint $\text{diag}(W^*) = 0$, expressed as

$$W^* = \frac{1}{1-p} \left( I - C \cdot (I \odot C)^{-1} \right), \text{ where } C = \left( R^T R + \frac{p}{1-p} I \odot R^T R \right)^{-1} \qquad (4)$$

While (3) remains valid and meaningful for $b > 0$, the solution for this case is not discussed in the original work. Moreover, the suggested hyperparameter choice $a \geq b$ assigns greater weight to the reconstruction loss of dropped interactions and is intended to enhance the model's ability to recover them. This design aligns with the MSE used during evaluation (Appendix G.2).

## 3 DECOUPLED EXPECTED QUADRATIC LOSS FOR LINEAR AUTOENCODERS

We first rewrite the EDLAE objective function in (3) into an expectation form. Let $\mathcal{B}$ denote the multivariate Bernoulli distribution of $\Delta$, then by the law of large numbers, (3) can be rewritten as

$$W^* = \text{argmin}_W \, l_{\mathcal{B}}(W), \quad \text{where}$$

$$l_{\mathcal{B}}(W) = \mathbb{E}_{\Delta \sim \mathcal{B}} \left[ \|A \odot (R - (\Delta \odot R)W)\|_F^2 \right] = \sum_{i=1}^{n} \mathbb{E}_{\Delta \sim \mathcal{B}} \left[ \|A_{*i} \odot (R_{*i} - (\Delta \odot R)W_{*i})\|_F^2 \right]$$

$$= \sum_{i=1}^{n} \mathbb{E}_{\Delta \sim \mathcal{B}} \left[ \|A^{(i)} R_{*i} - A^{(i)}(\Delta \odot R)W_{*i}\|_F^2 \right] \qquad (5)$$

Here we denote $A^{(i)} = \text{diagMat}(A_{*i})$. Note that (5) *decouples* the squared Frobenius norm over the columns of $W$. Since $R$ is constant while both $\Delta$ and $A^{(i)}$ are random, define $Y^{(i)} = A^{(i)}R$ and $X^{(i)} = A^{(i)}(\Delta \odot R)$, then both $X^{(i)}$ and $Y^{(i)}$ are random. If we further denote $\mathcal{D}^{(i)}$ as the distribution of the pair $(X^{(i)}, Y^{(i)})$, then the objective function in (5) can be written as

$$l_{\mathcal{B}}(W) = \sum_{i=1}^{n} \mathbb{E}_{(X^{(i)}, Y^{(i)}) \sim \mathcal{D}^{(i)}} \left[ \|Y^{(i)} - X^{(i)} W_{*i}\|_F^2 \right] \qquad (6)$$

where each column $W_{*i}$ is placed inside an *expected quadratic loss* $\mathbb{E}_{(X^{(i)}, Y^{(i)}) \sim \mathcal{D}^{(i)}} \left[ \|Y^{(i)} - X^{(i)} W_{*i}\|_F^2 \right]$. This formulation is general since each $\mathcal{D}^{(i)}$ can be any distribution, while (5) is a special case where $X^{(i)}$ and $Y^{(i)}$ follow distributions induced by applying random dropout to constants. We first derive the general closed-form solution of optimizing (6), then specialize it to EDLAE, and show that this reformulation simplifies the analysis and reveals a broader class of solutions for $b \geq 0$ compared Steck's original solution for $b = 0$ (4).

### 3.1 DECOUPLED EXPECTED QUADRATIC LOSS AND ITS CLOSED-FORM SOLUTION

We formally define (6) as follows:

**Definition 3.1.** Given a set of joint distributions $\mathcal{D} = \{\mathcal{D}^{(i)}\}_{i=1}^n$ over the pair $(X, Y)$, the **decoupled expected quadratic loss (DEQL)** is defined as

$$l_{\mathcal{D}}(W) = \sum_{i=1}^n h_{\mathcal{D}^{(i)}}^i(W_{*i}), \text{ where}$$

$$\begin{aligned} h_{\mathcal{D}^{(i)}}^i(W_{*i}) &= \mathbb{E}_{(X,Y)\sim\mathcal{D}^{(i)}} \left[\|Y_{*i} - XW_{*i}\|_F^2\right] \\ &= W_{*i}^T \mathbb{E}_{(X,Y)\sim\mathcal{D}^{(i)}} \left[X^T X\right] W_{*i} - 2W_{*i}^T \mathbb{E}_{(X,Y)\sim\mathcal{D}^{(i)}} \left[X^T Y_{*i}\right] + \mathbb{E}_{(X,Y)\sim\mathcal{D}^{(i)}} \left[Y_{*i}^T Y_{*i}\right] \end{aligned}$$
(7)

Note that each $h_{\mathcal{D}^{(i)}}^i$ is a quadratic function of $W_{*i}$. Since $\mathbb{E}_{(X,Y)\sim\mathcal{D}^{(i)}} \left[X^T X\right]$ is positive semi-definite (as $X^T X$ is a random matrix whose realizations are always positive semi-definite), $h_{\mathcal{D}^{(i)}}^i$ is convex for any $i$. Hence, let $W^* = \operatorname{argmin}_W l_{\mathcal{D}}(W)$, then $W_{*i}^* = \operatorname{argmin}_{W_{*i}} h_{\mathcal{D}^{(i)}}^i(W_{*i})$ for all $i$. Furthermore, if $\mathbb{E}_{(X,Y)\sim\mathcal{D}^{(i)}} \left[X^T X\right]$ is positive definite for all $i$, so that its inverse exists, then $W^*$ can be computed column-wise as

$$W_{*i}^* = \mathbb{E}_{(X,Y)\sim\mathcal{D}^{(i)}} \left[X^T X\right]^{-1} \mathbb{E}_{(X,Y)\sim\mathcal{D}^{(i)}} \left[X^T Y_{*i}\right] \text{ for } i = 1, 2, ..., n$$
(8)

As a remark, (7) can be viewed as a generalization of the *true risk* of multivariate linear regression in statistical learning theory Vapnik (1999): by taking $\mathcal{D} := \mathcal{D}^{(1)} = \mathcal{D}^{(2)} = ... = \mathcal{D}^{(n)}$, (7) reduces to $l_{\mathcal{D}}(W) = \mathbb{E}_{(X,Y)\sim\mathcal{D}} \left[\|Y - XW\|_F^2\right]$, where $\|Y - XW\|_F^2$ is a multivariate linear regression loss.

Moreover, the solution $W^*$ obtained from (8) is in general full rank. We further discuss the case of optimizing (7) under a low-rank constraint on $W$ in Appendix D. In this case, closed-form solutions exist if $\mathbb{E}_{(X,Y)\sim\mathcal{D}^{(i)}} \left[X^T X\right]$ is independent of $i$ for all $i$.

## 3.2 ADAPTATION TO EDLAE

This section shows how we derive the closed-form solutions of EDLAE (3) from DEQL (7), covering the case $b \geq 0$. Our results show that, in the $b = 0$ case, the solution is not unique: they shared the same off-diagonal but can have arbitrary diagonal, while Steck's solution (Steck, 2020) is one of them by choosing a zero diagonal. Furthermore, we establish that for every $b > 0$, a closed-form solution exists and is unique – a property that was not analyzed in Steck's work.

We first discuss the $b > 0$ case. Remember that (5) is a special case of (7) by taking $X = A^{(i)}(\Delta \odot R)$ and $Y_{*i} = A^{(i)} R_{*i}$. By (8), the solution of (5) is given by

$$W_{*i}^* = H^{(i)-1} v^{(i)} \text{ for } i = 1, 2, ..., n, \text{ where}$$
(9)

$$H^{(i)} = \mathbb{E}_{\Delta \sim \mathcal{B}} \left[(\Delta \odot R)^T A^{(i)2} (\Delta \odot R)\right], \; v^{(i)} = \mathbb{E}_{\Delta \sim \mathcal{B}} \left[(\Delta \odot R)^T A^{(i)2} R_{*i}\right]$$
(10)

The following lemma enables explicit computation of the expectations in (10):

**Lemma 3.2.** *The $H^{(i)}$ and $v^{(i)}$ in (10) can be expressed as $H^{(i)} = G^{(i)} \odot R^T R$ and $v^{(i)} = u^{(i)} \odot R^T R_{*i}$, where $G^{(i)} \in \mathbb{R}^{n \times n}$ and $u^{(i)} \in \mathbb{R}^n$ satisfy*

$$G_{kl}^{(i)} = \begin{cases} (1-p)b^2 & \text{if } k = l = i \\ (1-p)^2 b^2 & \text{if } k \neq l = i \text{ or } l \neq k = i \\ (1-p)pa^2 + (1-p)^2 b^2 & \text{if } k = l \neq i \\ (1-p)^2 pa^2 + (1-p)^3 b^2 & \text{if } i \neq k \neq l \neq i \end{cases}, \quad u_k^{(i)} = \begin{cases} (1-p)b^2 & \text{if } k = i \\ (1-p)pa^2 + (1-p)^2 b^2 & \text{if } k \neq i \end{cases}$$

*for $k, l \in \{1, 2, ..., n\}$.*

Furthermore, the computation of (9) involves $H^{(i)-1}$, which exists only if $H^{(i)}$ is invertible. The following theorem establishes sufficient conditions to ensure this property.

**Theorem 3.3.** *For any $a \geq 0, b > 0$ and $0 < p < 1$, $G^{(i)}$ is positive definite. Furthermore, $H^{(i)}$ is positive definite if $G^{(i)}$ is positive definite and no column of $R$ is a zero vector.*

A positive definite $H^{(i)}$ is always invertible. Hence, Theorem 3.3 implies that the closed-form solution by (9) holds for any $a \geq 0$ and $b > 0$. Notably, this includes the case $b > a$, which lies outside the original EDLAE range $a \geq b \geq 0$ (see Figure 2).

Now we discuss the case when $b = 0$. In this setting, both the $i$-th row and $i$-th column of $H^{(i)}$ are zero, and the $i$-th row of $v^{(i)}$ is also zero. Consequently, $H^{(i)}$ is singular and its inverse $H^{(i)^{-1}}$ does not exist, making (8) inapplicable for computing the optimal $W^*$.

To proceed, we define submatrices and subvectors using the subscript $-i$ notation: if $Q$ is an $n \times n$ matrix, then $Q_{-i}$ is a $(n-1) \times (n-1)$ matrix obtained by removing the $i$-th row and $i$-th column of $Q$; if $q$ is an $n$ dimensional vector, then $q_{-i}$ is an $n-1$ dimensional vector obtained by removing $q_i$ from $q$. Under this notation, we can write $H_{-i}^{(i)} = G^- \odot (R^T R)_{-i}$, where $G^-$ is an $(n-1) \times (n-1)$ matrix with diagonal elements $(1-p)pa^2$ and off-diagonal elements $(1-p)^2 pa^2$. It is easy to verify that $G^-$ is positive definite, hence $H_{-i}^{(i)}$ is positive definite. Likewise, $v_{-i}^{(i)} = u^- \odot (R^T R_{*i})_{-i}$, where $u^-$ is an $n-1$ dimensional vector with all elements being $(1-p)pa^2$.

Denote the vector $(W_{*i})_{-i}$ as $W_{*i,-i}$, then (5) can be written as

$$l_{\mathcal{B}}(W) = \sum_{i=1}^{n} W_{*i}^T H^{(i)} W_{*i} - 2W_{*i}^T v^{(i)} + \mathbb{E}_{\Delta \sim \mathcal{B}} \left[ R_{*i}^T A^{(i)^2} R_{*i} \right] \tag{11}$$

$$= \sum_{i=1}^{n} W_{*i,-i}^T H_{-i}^{(i)} W_{*i,-i} - 2W_{*i,-i}^T v_{-i}^{(i)} + \mathbb{E}_{\Delta \sim \mathcal{B}} \left[ R_{*i}^T A^{(i)^2} R_{*i} \right] \tag{12}$$

Therefore, the solution $W^* = \operatorname{argmin}_W l_{\mathcal{B}}(W)$ is expressed as

$$W_{*i,-i}^* = (H_{-i}^{(i)})^{-1} v_{-i}^{(i)} \text{ and } W_{ii}^* \in \mathbb{R} \quad \text{for } i = 1, 2, ..., n \tag{13}$$

The optimal $W^*$ given by (13) is not unique, but belongs to an infinite set of solutions that share the same off-diagonal entries while allowing arbitrary diagonal entries. The following theorem shows that Steck's solution (4) is a special case of (13), corresponding to the choice of zero diagonal.

**Theorem 3.4.** *Suppose no column of $R$ is a zero vector. If taking $W_{ii} = 0$ for all $i$ in (13), then (13) and (4) are equivalent.*

It is important to note that varying the diagonal of $W^*$ can lead to different performance on test data, and (13) does not provide theoretical guidance on which choice of diagonal elements gives the best performance. However, empirical results suggest that a $W^*$ with non-zero diagonal elements can outperform the zero-diagonal solution in certain cases (Moon et al., 2023).

### 3.3 Adding $L_2$ Regularizer or Zero-diagonal Constraint

In LAE-based recommender systems, $L_2$ regularizer and zero diagonal constraint are commonly applied to the objective function, as they are established techniques for improving test performance. This section discusses the closed-form solution of the optimization problem (5) when the $L_2$ regularizer or the zero-diagonal constraint is applied.

**Adding $L_2$ Regularizer**: Given $\lambda > 0$, (5) with $L_2$ regularizer is expressed as
$$W^* = \operatorname{argmin}_W l_{\mathcal{B}}(W) + \lambda \|W\|_F^2 \tag{14}$$

**Adding Zero-diagonal Constraint**: (5) with zero-diagonal constraint is expressed as
$$W^* = \operatorname{argmin}_W l_{\mathcal{B}}(W) \quad \text{s.t. diag}(W) = 0 \tag{15}$$

In these cases, the solution (9) is modified accordingly, as presented below.

**Proposition 3.5.** *(a) The solution of (14) is*

$$W_{*i}^* = \left( H^{(i)} + \lambda I \right)^{-1} v^{(i)} \quad \text{for } i = 1, 2, ..., n \tag{16}$$

*(b) The solution of (15) is*

$$W_{*i}^* = H^{(i)^{-1}} v^{(i)} - \frac{(H^{(i)^{-1}} v^{(i)})_i}{(H^{(i)^{-1}} l^{(i)})_i} H^{(i)^{-1}} l^{(i)} \quad \text{for } i = 1, 2, ..., n \tag{17}$$

*where $l^{(i)}$ is an $n$-dimensional vector with $l_i^{(i)} = 1$ and $l_j^{(i)} = 0$ for all $j \neq i$.*

## 4   AN EFFICIENT ALGORITHM FOR THE CLOSED-FORM SOLUTION

Recall from Theorem 3.3 that the optimal $W^*$ for the $b > 0$ case of EDLAE can be computed by (9). However, a major challenge is its high computational complexity: since $H^{(i)}$ differs for each $i$, computing each inverse $H^{(i)^{-1}}$ costs $O(n^3)$ (suppose we use *basic* matrix inverse algorithms, e.g., Cholesky decomposition (Krishnamoorthy & Menon, 2013)), resulting in a total cost of $O(n^4)$ for all $i$, which is computationally impractical.

In this section, we show the existence of a practical algorithm that reduces the overall complexity of computing (9) from $O(n^4)$ to $O(n^3)$. We prove this by explicitly constructing one using Miller's matrix inverse theorem:

**Theorem 4.1.** *((Miller, 1981)) Let $G$ and $G + Q$ be non-singular matrices. Suppose $Q$ is of rank $r$ and can be decomposed as $Q = E_1 + E_2 + ... + E_r$, where each $E_k$ is of rank 1, and $P_{k+1} = G + E_1 + E_2 + ... + E_k$ is non-singular for $k = 1, 2, ..., r$. Let $P_1 = G$, then*

$$P_{k+1}^{-1} = P_k^{-1} - \frac{1}{1 + \text{tr}\left(P_k^{-1} E_k\right)} P_k^{-1} E_k P_k^{-1}$$

In Lemma 3.2, we define $H^{(i)} = G^{(i)} \odot R^T R$, which can be decomposed as

$$H^{(i)} = G_0 \odot R^T R + G_1^{(i)} \odot R^T R + G_2^{(i)} \odot R^T R$$

where $G_0$ is a matrix with diagonal elements equal to $(1 - p)pa^2 + (1 - p)^2 b^2$ and off-diagonal elements equal to $(1 - p)^2 pa^2 + (1 - p)^3 b^2$; $G_1^{(i)}$ is a matrix with $(G_1^{(i)})_{ji} = -(1 - p)^2 p(a^2 - b^2)$ for $j \neq i$, $(G_1^{(i)})_{ii} = -(1 - p)p(a^2 - b^2)$, and all other elements zero; $G_2^{(i)}$ is a matrix with $(G_2^{(i)})_{ij} = -(1 - p)^2 p(a^2 - b^2)$ for $j \neq i$ and all other elements zero.

Denote $H_0 = G_0 \odot R^T R$, $E_1^{(i)} = G_1^{(i)} \odot R^T R$ and $E_2^{(i)} = G_2^{(i)} \odot R^T R$, then $H^{(i)} = H_0 + E_1^{(i)} + E_2^{(i)}$. Note that $H_0$ is positive definite and independent of $i$, $E_1^{(i)}$ is of rank 1 with only the $i$-th column being nonzero, and $E_2^{(i)}$ is of rank 1 with the $i$-th row (excluding $(E_2^{(i)})_{ii}$) being nonzero.

Applying Theorem 4.1, $H^{(i)^{-1}}$ can be computed with the following two steps:

$$H_+^{(i)^{-1}} = (H_0 + E_1^{(i)})^{-1} = H_0^{-1} - \frac{1}{1 + \text{tr}(H_0^{-1} E_1^{(i)})} H_0^{-1} E_1^{(i)} H_0^{-1} \tag{18}$$

$$H^{(i)^{-1}} = (H_0 + E_1^{(i)} + E_2^{(i)})^{-1} = H_+^{-1} - \frac{1}{1 + \text{tr}(H_+^{-1} E_2^{(i)})} H_+^{-1} E_2^{(i)} H_+^{-1} \tag{19}$$

Let $e_1^{(i)}$ be the $i$-th column of $E_1^{(i)}$, $e_2^{(i)^T}$ be the $i$-th row of $E_2^{(i)}$, then (18) and (19) can be simplified as

$$H_+^{(i)^{-1}} = H_0^{-1} - \frac{1}{1 + (H_0^{-1})_{i*} e_1^{(i)}} (H_0^{-1} e_1^{(i)})(H_0^{-1})_{i*} \tag{20}$$

$$H^{(i)^{-1}} = H_+^{(i)^{-1}} - \frac{1}{1 + e_2^{(i)^T} (H_+^{(i)^{-1}})_{*i}} (H_+^{(i)^{-1}})_{*i}(e_2^{(i)^T} H_+^{(i)^{-1}}) \tag{21}$$

Observe that, given $H_0^{-1}$, the computation of each $H^{(i)^{-1}}$ using (20) and (21) requires only $O(n^2)$ operations, resulting in a total cost of $O(n^3)$ for all $i$. This significantly reduces the original $O(n^4)$ complexity of computing (9).

Moreover, the computation can be further simplified by directly computing $H^{(i)^{-1}} v^{(i)}$ without explicitly forming $H^{(i)^{-1}}$. By (20) and (21),

$$H_+^{(i)^{-1}} v^{(i)} = H_0^{-1} v^{(i)} - \frac{1}{1 + (H_0^{-1})_{i*} e_1^{(i)}} (H_0^{-1} e_1^{(i)}) \left[(H_0^{-1})_{i*} v^{(i)}\right] \tag{22}$$

$$H^{(i)^{-1}} v^{(i)} = H_+^{(i)^{-1}} v^{(i)} - \frac{1}{1 + e_2^{(i)^T} (H_+^{(i)^{-1}})_{*i}} (H_+^{(i)^{-1}})_{*i} \left[(e_2^{(i)^T} H_+^{(i)^{-1}}) v^{(i)}\right] \tag{23}$$

in which the scalars $(H_0^{-1})_{i*}v^{(i)}$ and $(e_2^{(i)T}H_+^{(i)-1})v^{(i)}$ can be computed first. In (23), the $(H_+^{(i)-1})_{*i}$ term can be computed by (20),

$$(H_+^{(i)-1})_{*i} = (H_0^{-1})_{*i} - \frac{(H_0^{-1})_{ii}}{1 + (H_0^{-1})_{i*}e_1^{(i)}}(H_0^{-1}e_1^{(i)})$$

Denote $s = H_+^{(i)-1}v^{(i)}, t = (H_+^{(i)-1})_{*i}$. Let $U$ be an $n \times n$ matrix with diagonal elements equal to $(1-p)b^2$ and off-diagonal elements equal to $(1-p)pa^2 + (1-p)^2b^2$, let $G_1$ be an $n \times n$ matrix with diagonal elements $-(1-p)p(a^2 - b^2)$ and off-diagonal elements $-(1-p)^2p(a^2 - b^2)$, and let $G_2$ be an $n \times n$ matrix with zeros on the diagonal and off-diagonal elements $-(1-p)^2p(a^2 - b^2)$. Then we can summarize our computation as follows.

**Fast Algorithm for Computing (9)**: First, precompute these matrices

$$R^T R, \quad H_0^{-1} = \left(G_0 \odot R^T R\right)^{-1}, \quad [H_0^{-1}v^{(1)}, H_0^{-1}v^{(2)}, ..., H_0^{-1}v^{(n)}] = H_0^{-1}(U \odot R^T R),$$
$$[H_0^{-1}e_1^{(1)}, H_0^{-1}e_1^{(2)}, ..., H_0^{-1}e_1^{(n)}] = H_0^{-1}(G_1 \odot R^T R), \quad [e_2^{(1)}, e_2^{(2)}, ..., e_2^{(n)}] = G_2 \odot R^T R$$

Then for $i = 1, 2, ..., n$, compute each $W_{*i}^* = H^{(i)-1}v^{(i)}$ as follows:

$$r = H_0^{-1}v^{(i)}, \quad w = H_0^{-1}e_1^{(i)}$$
$$s = r - \frac{1}{1+w_i}r_i w, \quad t = (H_0^{-1})_{*i} - \frac{(H_0^{-1})_{ii}}{1+w_i}w$$
$$H^{(i)-1}v^{(i)} = s - \frac{1}{1 + e_2^{(i)T}t}(e_2^{(i)T}s)t$$

We now analyze the computational complexity of the above algorithm. Here we assume that matrix multiplication and inversion are implemented using *basic* linear algebra algorithms. In the precomputing stage, $R^T R$ costs $O(mn^2)$, $H_0^{-1}$ costs $O(n^3)$, $H_0^{-1}(U \odot R^T R)$ and $H_0^{-1}(G_1 \odot R^T R)$ cost $O(n^3)$, and $G_2 \odot R^T R$ costs $O(n^2)$. In the computing stage, each $W_{*i}^*$ is obtained via only vector-vector multiplications, with a complexity of $O(n)$, resulting in an overall complexity of $O(n^2)$ for the entire $W^*$. Therefore, the total complexity of this algorithm is $O\left(\max(m+n)n^2\right)$.

This complexity is the same as the closed-form solutions of EASE (Steck, 2019) and EDLAE (Steck, 2020), and the main bottleneck lies in the $O(n^3)$ complexity for matrix inverse. If using *more advanced* algorithms for matrix multiplication and inversion, the complexity of our Fast Algorithm (as well as EASE and EDLAE) can be in further reduced from $O(n^3)$ to $O(n^{2.376})$. We elaborate on this in Appendix C.

**Adapting the Algorithm to the $L_2$ regularizer and Zero-diagonal Constraint Cases**: In (16), note that $H^{(i)} + \lambda I = (H_0 + \lambda I) + E_1^{(i)} + E_2^{(i)}$, where $H_0 + \lambda I$ is independent of $i$. This means that we can compute (16) using the above algorithm by replacing $H_0$ with $H_0 + \lambda I$. To compute (17), we use the algorithm to process the two components $H^{(i)-1}v^{(i)}$ and $H^{(i)-1}l^{(i)}$ respectively: first compute $H^{(i)-1}v^{(i)}$, then replace $v^{(i)}$ with $l^{(i)}$ and compute $H^{(i)-1}l^{(i)}$.

## 5 EXPERIMENTS

This section provides experimental results comparing DEQL with state-of-the-art collaborative filtering models, including linear models and deep learning based models.

### 5.1 EXPERIMENTAL SET-UP

**Datasets**: We utilize two group of public datasets. Group 1 includes Games, Beauty, Gowalla-1, ML-20M, Netflix, and MSD (Steck, 2019; Ni et al., 2019; Seol et al., 2024). This group adopts *strong generalization* setting, where the dataset is split into training and test sets by users (i.e., the users in the two sets do not overlap). Group 2 includes Amazonbook, Yelp2018 and Gowalla He et al. (2020). This group adopts *weak generalization* setting, where the dataset is split into training and test sets by interactions (i.e., a user may appear in both sets). Among these datasets, Gowalla-1

is preprocessed from the raw Gowalla dataset by us, whereas all other datasets follow the default training/test splits used in prior work.

Group 1 is used to compare DEQL with other LAE-based models, while Group 2 is used to compare DEQL with both deep learning-based and LAE-based models. We adopt this setting to align with common practice in the literature, where LAE models are more commonly evaluated under strong generalization, while deep learning models are typically evaluated under weak generalization. Details of these datasets are provided in Table 1.

Table 1: Details of Datasets

| Datasets | Group 1 | | | | | | Group 2 | | |
|---|---|---|---|---|---|---|---|---|---|
| | Games | Beauty | Gowalla-1 | ML20M | Netflix | MSD | Amazon-Books | Yelp2018 | Gowalla |
| # items | 896 | 4,394 | 13,681 | 20,108 | 17,769 | 41,140 | 91,599 | 38,048 | 40,981 |
| # users | 1,006 | 17,971 | 29,243 | 136,677 | 463,435 | 571,353 | 52,643 | 31,668 | 29,858 |
| # interactions | 15,276 | 75,472 | 677,956 | 9,990,682 | 56,880,037 | 33,633,450 | 2,984,108 | 1,561,406 | 1,027,370 |
| density | 1.69% | 0.10% | 0.17% | 0.36% | 0.69% | 0.14% | 0.06% | 0.13% | 0.08% |
| item-user ratio | 0.89 | 0.24 | 0.47 | 0.15 | 0.04 | 0.07 | 1.74 | 1.20 | 1.37 |

* density = # interactions / (# users × # items),  item-user ratio = # items / # users

**Baseline models and Evaluation Metrics**: We compare DEQL with the following state-of-the-art LAE-based models: EASE (Steck, 2019), DLAE (Steck, 2020), EDLAE (Steck, 2020), ELSA (Vančura et al., 2022), as well as the following recent deep learning-based models: PinSage (Ying et al., 2018), LightGCN (He et al., 2020), DGCF (Wang et al., 2020), SimpleX (Mao et al., 2021), SGL-ED (Wu et al., 2021) and SSM (Wu et al., 2024a). We evaluate model performance using widely adopted ranking metrics: Recall@20 (R@20) and NDCG@20 (N@20).

**Different versions of DEQL Models**: we test different versions of DEQL, distinguished as follows:

- DEQL(plain): the plain DEQL with $b > 0$, computed by (9).

- DEQL(L2): DEQL with $b > 0$ and $L_2$ regularization, computed by combining (9) and (16).

- DEQL(L2+zero-diag): DEQL with $b > 0$, $L_2$ regularization, and a zero-diagonal, computed by combining (9), (16) and (17).

Note that all baseline LAE models (EASE, EDLAE, DLAE and ELSA) in our experiments are equipped with $L_2$ regularization, consistent with their original papers [1]. Therefore, DEQL(L2) and DEQL(L2+zero-diag) provide a fair comparison with these baselines, while DEQL(plain) does not.

Moreover, the EDLAE baseline corresponds to DEQL with $b = 0$ and a zero-diagonal (see Theorem 3.4). Hence, in our experiments, DEQL refers only to the case $b > 0$.

**Hyperparameter Tuning**: We perform a grid search to tune the DEQL hyperparameters $a, b, p$. Note that for the DEQL objective (5) (as well as the EDLAE objective (3)), if both $a$ and $b$ are scaled by a constant $\alpha > 0$, the objective is scaled by $\alpha$, while solution $W^*$ remains unchanged. That is, $W^*$ is scalar invariant and depends solely on the ratio $b/a$. Hence, we fix $a = 1$ and search $b$ over the range [0.1,0.25,...,2.0]. The $L_2$ regularization coefficient $\lambda$ is searched over [10.0,20.0,...50,100.0,300.0,500.0], and the dropout rate $p$ is varied across [0.1,0.2,...,0.5,0.8].

**Hardware and Reproducibility**: All experiments are conducted on a Linux server equipped with 500 GB of memory, four NVIDIA 3090 GPUs, and a 96-core Intel(R) Xeon(R) Platinum 8268 CPU @ 2.90GHz. Our code is available at `https://github.com/coderaBruce/DEQL`.

## 5.2 MODEL PERFORMANCE EVALUATION

Table 2 compares DEQL with LAE-based models under the strong generalization setting. The results show that DEQL (plain) does not outperform the baselines, likely due to the absence of $L_2$ regularization, which prevents a fair comparison. However, DEQL(L2) and DEQL(L2+zero-diag) outperform the baselines by a small margin. These results indicate that:

1. The original EDLAE solution Steck (2020), obtained by setting $b = 0$ with a zero diagonal, is not optimal. Solutions in the $b > 0$ region, obtained via DEQL, can achieve higher test performance.

---

[1]For EDLAE, the author reports using an $L_2$ regularizer in their experimental set-up (Steck, 2020).

2. $L_2$ regularization is essential for improving performance, as DEQL(L2) and DEQL(L2+zero-diag) significantly outperform DEQL (plain).

3. The zero-diagonal constraint does not necessarily lead to better performance, since DEQL(L2) frequently outperforms DEQL(L2+zero-diag). In fact, the optimal DEQL(L2) solutions exhibit relatively small diagonal entries, as shown in Appendix F.2. This evidence supports the claim in Moon et al. (2023) that relaxing the zero-diagonal constraint in LAEs can improve performance.

Moreover, despite the relatively small performance gains of DEQL(L2) and DEQL(L2+zero-diag) in Table 2, statistical significance tests (see Appendix F.1) confirm that these improvements are stable and unlikely to result from statistical randomness.

Table 3 compares DEQL with deep learning-based and LAE-based models under the weak generalization setting. The result show that DEQL(L2) achieve the best performance on Amazonbook, outperforming competitors by up to 27% and 34% in R@20 and N@20, respectively, and surpasses most models on Yelp2018 and Gowalla.

Table 2: Performance comparison between DEQL and other LAE-based models under the strong generalization setting. The best results are highlighted in bold.

| Model | Games | | Beauty | | Gowalla-1 | | ML20M | | Netflix | | MSD | |
|---|---|---|---|---|---|---|---|---|---|---|---|---|
| | R@20 | N@20 | R@20 | N@20 | R@20 | N@20 | R@20 | N@20 | R@20 | N@20 | R@20 | N@20 |
| DLAE | 0.2771 | 0.1664 | 0.1329 | 0.0886 | 0.2143 | 0.1916 | 0.3924 | 0.3409 | 0.3620 | 0.3395 | 0.3290 | 0.3210 |
| EASE | 0.2733 | 0.1640 | 0.1323 | 0.0875 | 0.2230 | 0.1988 | 0.3905 | 0.3390 | 0.3618 | 0.3388 | 0.3332 | 0.3261 |
| EDLAE | 0.2851 | 0.1681 | 0.1324 | 0.0850 | 0.2268 | 0.2012 | 0.3925 | 0.3421 | 0.3656 | 0.3427 | 0.3336 | 0.3258 |
| ELSA | 0.2734 | 0.1658 | 0.1263 | 0.0763 | 0.2255 | 0.1960 | 0.3919 | 0.3386 | 0.3625 | 0.3372 | 0.3256 | 0.3144 |
| DEQL(plain) | 0.2524 | 0.1565 | 0.1093 | 0.0670 | 0.2149 | 0.1909 | 0.3844 | 0.3347 | 0.3606 | 0.3382 | 0.3329 | 0.3256 |
| DEQL(L2+zero-diag) | 0.2872 | 0.1704 | 0.1388 | **0.0898** | 0.2278 | 0.2027 | 0.3934 | **0.3429** | 0.3656 | 0.3423 | **0.3344** | **0.3268** |
| DEQL(L2) | **0.2998** | **0.1842** | **0.1391** | 0.0881 | **0.2288** | **0.2033** | **0.3934** | 0.3426 | **0.3658** | **0.3428** | 0.3340 | 0.3265 |

Table 3: Performance comparison between DEQL and deep learning–based and LAE-based models under the weak generalization setting.

| Model | Amazon-Books | | Yelp2018 | | Gowalla | |
|---|---|---|---|---|---|---|
| | R@20 | N@20 | R@20 | N@20 | R@20 | N@20 |
| **Deep learning based models** | | | | | | |
| PinSage | 0.0282 | 0.0219 | 0.0471 | 0.0393 | 0.1380 | 0.1196 |
| LightGCN | 0.0411 | 0.0315 | 0.0649 | 0.0530 | 0.1830 | 0.1554 |
| DGCF | 0.0422 | 0.0324 | 0.0654 | 0.0534 | 0.1842 | 0.1561 |
| SGL-ED | 0.0478 | 0.0379 | 0.0675 | 0.0555 | – | – |
| SimpleX | 0.0583 | 0.0468 | 0.0701 | 0.0575 | 0.1872 | 0.1557 |
| SSM (MF) | 0.0473 | 0.0367 | 0.0509 | 0.0404 | 0.1231 | 0.0878 |
| SSM (GNN) | 0.0590 | 0.0459 | **0.0737** | **0.0609** | **0.1869** | **0.1571** |
| **LAE-based models** | | | | | | |
| DLAE | **0.0751** | 0.0610 | 0.0678 | 0.0570 | 0.1839 | 0.1533 |
| EASE | 0.0710 | 0.0566 | 0.0657 | 0.0552 | 0.1765 | 0.1467 |
| EDLAE | 0.0711 | 0.0566 | 0.0673 | 0.0565 | 0.1844 | 0.1539 |
| ELSA | 0.0719 | 0.0594 | 0.0629 | 0.0541 | 0.1755 | 0.1490 |
| DEQL(plain) | 0.0695 | 0.0537 | 0.0647 | 0.0543 | 0.1749 | 0.1453 |
| DEQL(L2+zero-diag) | 0.0711 | 0.0567 | 0.0672 | 0.0565 | 0.1844 | 0.1539 |
| DEQL(L2) | **0.0751** | **0.0613** | 0.0685 | 0.0576 | 0.1845 | 0.1540 |

## 5.3 THE IMPACT OF $b$ ON MODEL PERFORMANCE

In this section, we conduct a sensitivity analysis on $b$ to investigate how different values of $b$ affect model performance.

We evaluate the effect of different $b$ across seven benchmark datasets, as shown in Figure 1. The first three datasets use weak generalization, while the last four datasets use strong generalization. For each dataset, we set $a = 1$ and vary $b$ from 0 to 2.0, and report their Recall@20 and NDCG@20. Note that for $b = 0$, the solution is obtained from (4); and for $b > 0$, the solution is obtained from (9), whose existence is guaranteed by Theorem 3.3.

On datasets *ML-20M*, *Games*, *Netflix*, and *MSD*, we observe a clear and consistent pattern: performance first improves when increasing $b$ from 0, reaches its peak *before* the $b/a$ ratio exceeds 1, and then gradually decreases as $b$ becomes too large. This behavior directly demonstrates that the $b = 0$ choice in the original EDLAE does not necessarily yield the best performance, and that models obtained with $b > 0$ using DEQL can achieve superior results.

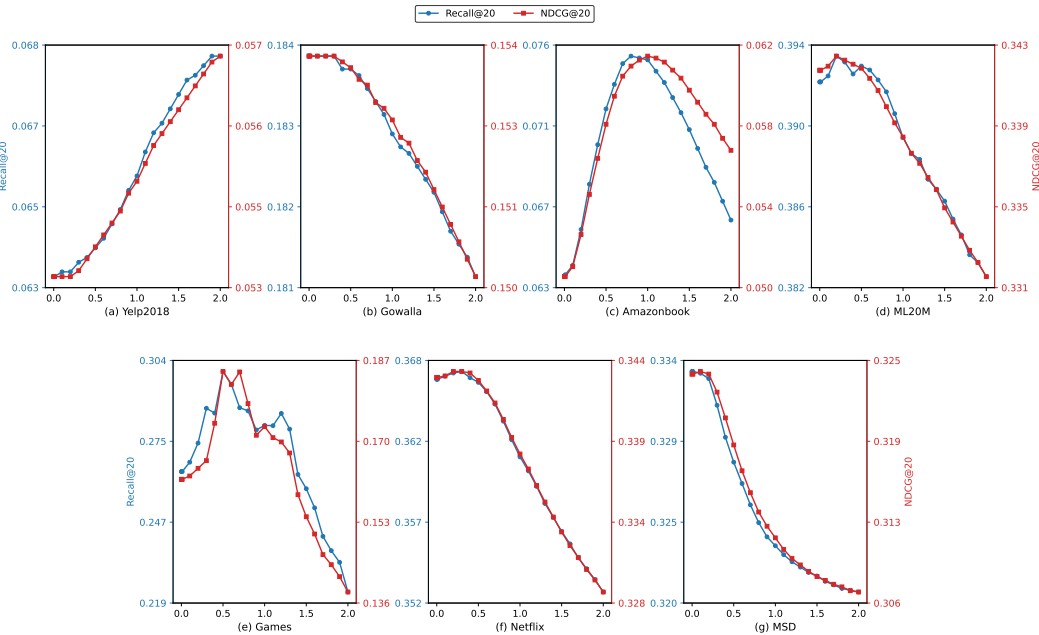

Figure 1: Sensitivity on $b/a$ ratio across different datasets

Interestingly, we observe a markedly different pattern on *Yelp2018* and *AmazonBook*: their optimal $b/a$ ratios approach 1 or even *exceed* 1 (as in Yelp). In the original EDLAE formulation, $a \geq b$ promotes reconstruction of *dropped* items through cross-item learning, with $b$ typically fixed to zero. The regime of $b > a$ has rarely been explored, as it instead trains the model to use the *remained* items to better predict other remained ones within the same set. Although $a > b$ appears to be the intuitive choice, our results show that $b > a$ can yield superior performance on certain datasets, suggesting that emphasizing dropped entries is not always beneficial.

Several factors may jointly contribute to this issue. We hypothesize that a key factor is the large **item-user ratio** and the resulting unreliable cross-item correlations. When the number of items greatly exceeds the number of users, as in *AmazonBook* and *Yelp2018*, the interaction matrix becomes extremely sparse. In such settings, cross-item correlations are weak or noisy, since most items receive very few interactions, providing insufficient data to estimate reliable item-item relationships for collaborative filtering (Nikolakopoulos et al., 2021). Under this regime, training with $a > b$ forces the model to rely more heavily on unreliable cross-item correlations, thereby degrading performance. In contrast, setting $b > a$ shifts the objective toward reconstructing the remained items themselves rather than the dropped ones, thereby stabilizing training through stronger self-association signals. This effect is reflected in the larger diagonal magnitudes of the learned weight matrices $W$, indicating increased reliance on identity-like mappings (Figure 3).

## 6    CONCLUSION

This paper aims to advance the EDLAE recommender system by extending its closed-form solution to a broader range of hyperparameter choices, and develop an efficient algorithm to compute these solutions. We first generalize the EDLAE objective function into DEQL, derive its closed-form solutions, and then apply them back to EDLAE. DEQL extends the original EDLAE solution for $b = 0$ to a wider range $b \geq 0$, enabling exploration of a larger solution space. To address the high computational complexity of solutions for $b > 0$, we develop an efficient algorithm based on Miller's matrix inverse theorem, reducing the complexity from $O(n^4)$ to $O(n^3)$.

Experimental results demonstrate that most solutions for $b > 0$ outperform the $b = 0$ baseline, showing that DEQL enables the discovery of models with better testing performance. Furthermore, on certain datasets, the optimal $b$ even lies in the regime $b > a$. To the best of our knowledge, this is the first empirical evidence showing that the commonly assumed EDLAE constraint $a \geq b$ is not universally optimal. Finally, we note that DEQL is a general loss function that may inspire the construction of other specialized objectives for LAE models.

ACKNOWLEDGMENTS

This research was partially supported by NSF grant IIS 2142675. We thank all reviewers for their insightful comments.

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

## A  ILLUSTRATION OF THE DEQL FRAMEWORK

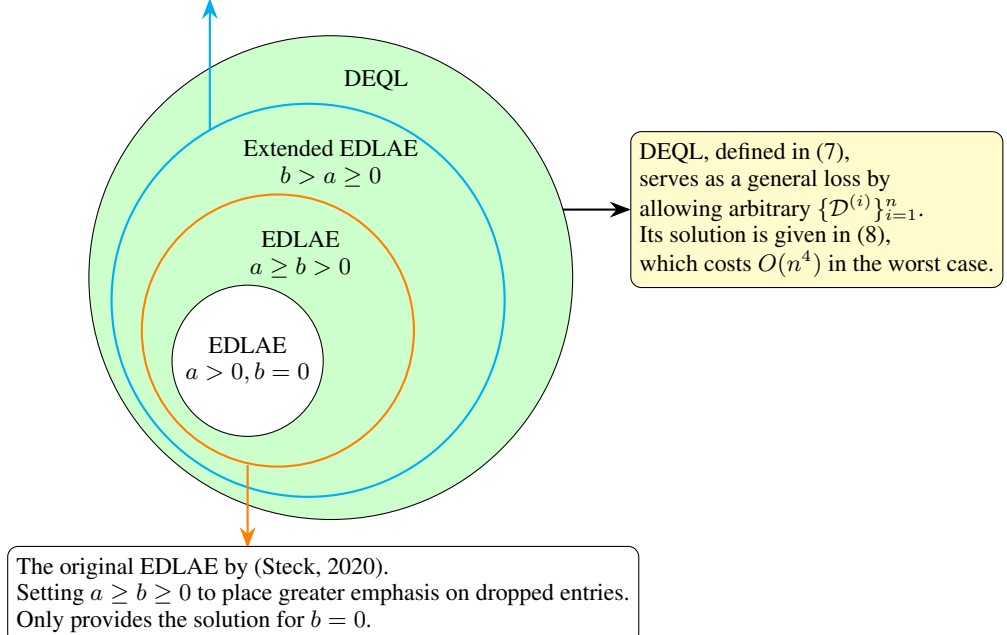

Figure 2: Comparison of the closed-form solution sets of DEQL and EDLAE. The white region represents the set of original EDLAE solution, and the green region represents the remaining solutions covered by DEQL. The orange circle marks solutions derived from a original EDLAE loss, whereas the cyan circle marks solutions obtained from the *extended* EDLAE loss (with hyperparameter choices $b > a$, which go beyond the original EDLAE constraints but still yield valid solutions).

## B  MATHEMATICAL PROOFS

*Proof of Lemma 3.2*: $H^{(i)}$ can be computed as follows. For any $k, l$,

$$H_{kl}^{(i)} = \mathbb{E}_\Delta[(\Delta \odot R)_{*k}^T A^{(i)^2}(\Delta \odot R)_{*l}] = \mathbb{E}_\Delta[\sum_{s=1}^m \Delta_{sk} R_{sk} A_{ss}^{(i)^2} \Delta_{sl} R_{sl}]$$

$$= \sum_{s=1}^m \mathbb{E}_\Delta[\Delta_{sk} R_{sk} A_{ss}^{(i)^2} \Delta_{sl} R_{sl}] = \sum_{s=1}^m \mathbb{E}_\Delta[\Delta_{sk} \Delta_{sl} A_{ss}^{(i)^2}] R_{sk} R_{sl} \qquad (24)$$

Note that $A_{ss}^{(i)} = A_{si}$, which depends on $\Delta_{si}$. Since we assume each $\Delta_{ij}$ is an i.i.d. Bernoulli random variable, $\mathbb{E}_\Delta[\Delta_{sk} \Delta_{sl} A_{si}^2]$ is independent of $s$. Thus we can let a $z$ be a specific value of $s$ and rewrite (24) as

$$H_{kl}^{(i)} = \mathbb{E}_\Delta[\Delta_{zk} \Delta_{zl} A_{zi}^2] \sum_{s=1}^m R_{sk} R_{sl} = \mathbb{E}_\Delta[\Delta_{zk} \Delta_{zl} A_{zi}^2] R_{*k}^T R_{*l}$$

Define $G^{(i)} \in \mathbb{R}^{n \times n}$ where $G_{kl}^{(i)} = \mathbb{E}_\Delta[\Delta_{zk} \Delta_{zl} A_{zi}^2]$, then $H^{(i)} = G^{(i)} \odot R^T R$. $G^{(i)}$ can be computed as follows: Given $i$, for any $k, l$,

$$\Delta_{zk} \Delta_{zl} A_{zi}^2 = \begin{cases} a^2 & \text{if } \Delta_{zk} = 1 \text{ and } \Delta_{zl} = 1 \text{ and } \Delta_{zi} = 0 \\ b^2 & \text{if } \Delta_{zk} = 1 \text{ and } \Delta_{zl} = 1 \text{ and } \Delta_{zi} = 1 \\ 0 & \text{otherwise} \end{cases}$$

Since

$$P\left(\Delta_{zk} = 1 \text{ and } \Delta_{zl} = 1 \text{ and } \Delta_{zi} = 0\right) = \begin{cases} (1-p)p & \text{if } k = l \neq i \\ (1-p)^2 p & \text{if } i \neq k \neq l \neq i \end{cases}$$

$$P\left(\Delta_{zk} = 1 \text{ and } \Delta_{zl} = 1 \text{ and } \Delta_{zi} = 1\right) = \begin{cases} 1-p & \text{if } k = l = i \\ (1-p)^2 & \text{if } k = l \neq i \text{ or } k \neq l = i \text{ or } l \neq k = i \\ (1-p)^3 & \text{if } i \neq k \neq l \neq i \end{cases}$$

we have

$$G_{kl}^{(i)} = \mathbb{E}_\Delta[\Delta_{zk}\Delta_{zl}A_{zi}^2] = \begin{cases} (1-p)b^2 & \text{if } k = l = i \\ (1-p)^2 b^2 & \text{if } k \neq l = i \text{ or } l \neq k = i \\ (1-p)pa^2 + (1-p)^2 b^2 & \text{if } k = l \neq i \\ (1-p)^2 pa^2 + (1-p)^3 b^2 & \text{if } i \neq k \neq l \neq i \end{cases}$$

On the other hand, $v^{(i)}$ can be computed as follows. For any $k$,

$$v_k^{(i)} = \mathbb{E}_\Delta[(\Delta \odot R)_{*k}^T A^{(i)^2} R_{*i}] = \mathbb{E}_\Delta[\sum_{s=1}^m \Delta_{sk} R_{sk} A_{ss}^{(i)^2} R_{si}] = \sum_{s=1}^m \mathbb{E}_\Delta[\Delta_{sk} A_{si}^2] R_{sk} R_{si}$$

$$= \mathbb{E}_\Delta[\Delta_{zk} A_{zi}^2] R_{*k}^T R_{*i}$$

Define $u^{(i)} \in \mathbb{R}^n$ where $u_k^{(i)} = \mathbb{E}_\Delta[\Delta_{zk} A_{zi}^2]$, then we can write $v^{(i)} = u^{(i)} \odot R^T R_{*i}$. $u^{(i)}$ can be computed as follows: Given any $k$,

$$u_k^{(i)} = \mathbb{E}_\Delta[\Delta_{zk} A_{zi}^2] = \begin{cases} (1-p)b^2 & \text{if } k = i \\ (1-p)pa^2 + (1-p)^2 b^2 & \text{if } k \neq i \end{cases}$$

$\square$

*Proof of Theorem 3.3*: Observe that $G^{(i)}$ can be decomposed as the sum of two matrices:

$$G^{(i)} = (1-p)b^2 M^{(i)} + (1-p)pa^2 N^{(i)}$$

where

$$M_{kl}^{(i)} = \begin{cases} 1 & \text{if } k = l = i \\ 1-p & \text{if } k \neq l = i \text{ or } l \neq k = i \text{ or } k = l \neq i \\ (1-p)^2 & \text{if } i \neq k \neq l \neq i \end{cases}$$

$$N_{kl}^{(i)} = \begin{cases} 0 & \text{if } k = l = i \text{ or } k \neq l = i \text{ or } l \neq k = i \\ 1 & \text{if } k = l \neq i \\ 1-p & \text{if } i \neq k \neq l \neq i \end{cases}$$

for $k, l \in \{1, 2, ..., n\}$.

We can show that for any $i$, $M^{(i)}$ is positive definite and $N^{(i)}$ is positive semi-definite: Let $x = [x_1, x_2, ..., x_n]^T \in \mathbb{R}^n$,

$$x^T M^{(i)} x = \left(x_i + (1-p)\sum_{\substack{j=1 \\ j \neq i}}^n x_j\right)^2 + p(1-p)\left(\sum_{\substack{j=1 \\ j \neq i}}^n x_j^2\right) > 0 \quad \text{for any } x \neq 0$$

$$x^T N^{(i)} x = (1-p)\left(\sum_{\substack{j=1 \\ j \neq i}}^n x_j\right)^2 + p\left(\sum_{\substack{j=1 \\ j \neq i}}^n x_j^2\right) \geq 0 \quad \text{for any } x$$

Hence, $G^{(i)}$ is positive definite if $a \geq 0, b > 0$ and $0 < p < 1$.

$R^T R$ is a positive semi-definite matrix. If no column of $R$ is a zero vector, then all diagonal elements of $R^T R$ are positive. By the Schur product theorem (Theorem 7.5.3 (b), (Horn & Johnson, 2012)), if $G^{(i)}$ is positive definite and the diagonal elements of $R^T R$ are all positive, then $H^{(i)} = G^{(i)} \odot R^T R$ is positive definite.

□

*Proof of Theorem 3.4*: Since the $W^*$ in (13) has zero diagonal, we only need to verify the equivalence of non-diagonal elements. Let us write $(C_{*i})_{-i}$ as $C_{*i,-i}$, then $W_{*i,-i}$ by (4) is expressed as

$$W_{*i,-i} = -\frac{1}{1-p}\frac{1}{C_{ii}}C_{*i,-i}$$

Thus, our goal is to prove

$$(H_{-i}^{(i)})^{-1}v_{-i}^{(i)} = -\frac{1}{1-p}\frac{1}{C_{ii}}C_{*i,-i} \text{ for any } i \in \{1, 2, ..., n\} \tag{25}$$

To show this, first,

$$
\begin{aligned}
(H_{-i}^{(i)})^{-1}v_{-i}^{(i)} &= \left(G^- \odot (R^T R)_{-i}\right)^{-1} \cdot (1-p)pa^2(R^T R_{*i})_{-i} \\
&= \left(\frac{1}{(1-p)pa^2}G^- \odot (R^T R)_{-i}\right)^{-1}(R^T R_{*i})_{-i} \\
&= \frac{1}{1-p}\left((R^T R)_{-i} + \frac{p}{1-p}I \odot (R^T R)_{-i}\right)^{-1}(R^T R_{*i})_{-i} \tag{26}
\end{aligned}
$$

Next, remember that $C^{-1} = R^T R + \frac{p}{1-p}I \odot R^T R$. Since no column of $R$ is a zero vector, $I \odot R^T R$ is positive definite, thus $C^{-1}$ is positive definite. By the properties of matrix inverse, for an invertible matrix $E$, if we swap the $i$-th and $j$-th rows (or columns) of $E$ and get $E'$, then $E'^{-1}$ is equivalent to the matrix formed by swapping the $i$-th and $j$-th columns (or rows) of $E^{-1}$. Therefore, suppose $(C^{-1})^{\langle i\rangle}$ is obtained from $C^{-1}$ by first swapping the $k$th row with the $(k+1)$th row for $k = i, i+1, ..., n-1$ sequentially, then swapping the $k$th column with the $(k+1)$th column for $k = i, i+1, ..., n-1$ sequentially. We have

$$(C^{-1})^{\langle i\rangle} = \begin{bmatrix} (R^T R)_{-i} + \frac{p}{1-p}I \odot (R^T R)_{-i} & (R^T R_{*i})_{-i} \\ (R^T R_{*i})_{-i}^T & \frac{1}{1-p}(R^T R)_{ii} \end{bmatrix}$$

and

$$\left((C^{-1})^{\langle i\rangle}\right)^{-1} = C^{\langle i\rangle} = \begin{bmatrix} M & C_{*i,-i} \\ C_{*i,-i}^T & C_{ii} \end{bmatrix}$$

where $M$ is an $(n-1) \times (n-1)$ matrix that we are not interested in.

By the symmetric block matrix inverse (0.7.3, (Horn & Johnson, 2012)), we know that

$$\begin{bmatrix} A & B^T \\ B & D \end{bmatrix}^{-1} = \begin{bmatrix} A^{-1} + A^{-1}B^T S^{-1}BA^{-1} & -A^{-1}B^T S^{-1} \\ -S^{-1}BA^{-1} & S^{-1} \end{bmatrix}$$

where $S = D - BA^{-1}B^T$.

Let $A = (R^T R)_{-i} + \frac{p}{1-p}(I \odot (R^T R)_{-i})$, $B^T = (R^T R_{*i})_{-i}$ and $S^{-1} = C_{ii}$, we have

$$C_{*i,-i} = -A^{-1}B^T S^{-1} = -\left((R^T R)_{-i} + \frac{p}{1-p}I \odot (R^T R)_{-i}\right)^{-1}(R^T R_{*i})_{-i} \cdot C_{ii} \tag{27}$$

Combining (27) and (26), we get (25), thereby completing the proof.

□

*Proof of Proposition 3.5*:

(a) Similar to (11), we can expand objective function of (14) as

$$l_{\mathcal{B}}(W) + \lambda\|W\|_F^2 = \sum_{i=1}^{n} h_{\mathcal{B}}^{(i)}(W_{*i}), \quad \text{where}$$

$$h_{\mathcal{B}}^{(i)}(W_{*i}) = W_{*i}^T H^{(i)} W_{*i} - 2W_{*i}^T v^{(i)} + \lambda\|W_{*i}\|_F^2 + \mathbb{E}_{\Delta\sim\mathcal{B}}\left[R_{*i}^T A^{(i)^2} R_{*i}\right]$$

Hence, $\left[\frac{\partial h_{\mathcal{B}}^{(i)}}{\partial W_{*i}}\right]^T = 2H^{(i)}W_{*i} - 2v^{(i)} + 2\lambda W_{*i}$, and the solution of $\left[\frac{\partial h_{\mathcal{B}}^{(i)}}{\partial W_{*i}}\right]^T = 0$ becomes

$$W_{*i}^* = \left(H^{(i)} + \lambda I\right)^{-1} v^{(i)} \tag{28}$$

The optimal $W^*$ is obtained by solving $W_{*i}^*$ via (28) for all $i$. The solution is unique since each $h_{\mathcal{B}}^{(i)}$ is strictly convex.

(b) (15) is equivalent to solving for the stationary points of the following a Lagrangian function

$$\mathcal{L}(W, \mu) = l_{\mathcal{B}}(W) + \mu^T \text{diag}(W)$$

where $\mu \in \mathbb{R}^n$. Since

$$\mathcal{L}(W, \mu) = \sum_{i=1}^{n} \bar{h}_{\mathcal{B}}^{(i)}(W_{*i}, \mu_i), \quad \text{where}$$

$$\bar{h}_{\mathcal{B}}^{(i)}(W_{*i}, \mu_i) = W_{*i}^T H^{(i)} W_{*i} - 2W_{*i}^T v^{(i)} + \mu_i W_{ii} + \mathbb{E}_{\Delta\sim\mathcal{B}}\left[R_{*i}^T A^{(i)^2} R_{*i}\right]$$

the solution $(W, \mu)$ of the system of equations

$$\left[\frac{\partial\mathcal{L}}{\partial W}\right]^T = \left[\left[\frac{\partial\mathcal{L}}{\partial W_{*1}}\right]^T, \left[\frac{\partial\mathcal{L}}{\partial W_{*2}}\right]^T, ..., \left[\frac{\partial\mathcal{L}}{\partial W_{*n}}\right]^T\right] = \left[\left[\frac{\partial\bar{h}_{\mathcal{B}}^{(1)}}{\partial W_{*1}}\right]^T, \left[\frac{\partial\bar{h}_{\mathcal{B}}^{(2)}}{\partial W_{*2}}\right]^T, ..., \left[\frac{\partial\bar{h}_{\mathcal{B}}^{(n)}}{\partial W_{*n}}\right]^T\right] = 0$$

$$\left[\frac{\partial\mathcal{L}}{\partial\mu}\right]^T = \left[\frac{\partial\mathcal{L}}{\partial\mu_1}, \frac{\partial\mathcal{L}}{\partial\mu_2}, ..., \frac{\partial\mathcal{L}}{\partial\mu_n}\right]^T = \left[\frac{\partial\bar{h}_{\mathcal{B}}^{(1)}}{\partial\mu_1}, \frac{\partial\bar{h}_{\mathcal{B}}^{(2)}}{\partial\mu_2}, ..., \frac{\partial\bar{h}_{\mathcal{B}}^{(n)}}{\partial\mu_n}\right]^T = 0$$

is given by taking

$$\left[\frac{\partial\bar{h}_{\mathcal{B}}^{(i)}}{\partial W_{*i}}\right]^T = 2H^{(i)}W_{*i} - 2v^{(i)} + \mu_i l^{(i)} = 0 \tag{29}$$

$$\frac{\partial\bar{h}_{\mathcal{B}}^{(i)}}{\partial\mu_i} = W_{ii} = 0 \tag{30}$$

for $i = 1, 2, ..., n$. Solving (29), we get

$$W_{*i} = H^{(i)^{-1}}(v_i - \frac{1}{2}\mu_i l^{(i)}) \tag{31}$$

Combining (30) and (31), we have

$$W_{ii} = (H^{(i)^{-1}}v^{(i)})_i - \frac{1}{2}\mu_i(H^{(i)^{-1}}l^{(i)})_i = 0 \implies \mu_i = 2\frac{(H^{(i)^{-1}}v^{(i)})_i}{(H^{(i)^{-1}}l^{(i)})_i} \tag{32}$$

Finally, plugging (33) into (31), we get the solution of $W^*$: For any $i$,

$$W_{*i}^* = H^{(i)^{-1}}(v^{(i)} - \frac{(H^{(i)^{-1}}v^{(i)})_i}{(H^{(i)^{-1}}l^{(i)})_i}l^{(i)}) = H^{(i)^{-1}}v^{(i)} - \frac{(H^{(i)^{-1}}v^{(i)})_i}{(H^{(i)^{-1}}l^{(i)})_i}H^{(i)^{-1}}l^{(i)} \tag{33}$$

The solution (33) is unique. By *second order sufficiency conditions* (Section 11.5, (Luenberger & Ye, 2008)), one can show that any $W^*$ that minimizes $\mathcal{L}(W, \mu)$ is a strict local minimizer. Thus, the solution (33) gives the global minimizer.

$\square$

## C    Improved Complexity for the Fast Algorithm

As discussed in Section 4, when using *basic* algorithms for matrix multiplication and inversion, our Fast Algorithm a computational cost of $O(\max(m + n)n^2)$. The main bottleneck lies in the precomputing stage: the $m \times n$ product $R^T R$ costs $O(mn^2)$; the $n \times n$ inversion $H_0^{-1}$ costs $O(n^3)$; and multiplying $H_0^{-1}$ with $n \times n$ matrices $U \odot R^T R$ and $G_1 \odot R^T R$ costs $O(n^3)$. The following corollary shows that these costs can be reduced when *advanced* matrix multiplication and inversion algorithms are applied.

**Corollary C.1.** *(a) If $R$ is sparse, contains only integer elements, and has $k$ nonzero elements $(\max(m, n) < k < mn)$, then $R^T R$ can be computed with complexity $O((2k + n^2)^{1.346})$.*

*(b) The cost of computing $H_0^{-1}$ can be reduced to $O(n^{2.376})$, but cannot be improved beyond $\Omega(n^2 \log n)$.*

*(c) The cost of computing $H_0^{-1}(U \odot R^T R)$ and $H_0^{-1}(U \odot R^T R)$ can each be reduced to $O(n^{2.376})$, but cannot be improved beyond $\Omega(n^2 \log n)$.*

*Proof*:

(a) By the analysis of (Abboud et al., 2024), consider the multiplication of sparse integer matrices $A^{x \times y}$ and $B^{y \times z}$. Let $m_{\text{in}}$ be the total number of nonzeros in the inputs $A$ and $B$, and let $m_{\text{out}}$ be the number of nonzeros in the output $AB$. If $m_{\text{in}} \geq \max(x, y, z)$, then the matrix multiplication can be computed with complexity $O((m_{\text{in}} + m_{\text{out}})^{1.346})$.

For computing $R^T R$, we have $m_{\text{in}} = 2k$ and $m_{\text{out}} \leq n^2$, so the total complexity is $O((2k + n^2)^{1.346})$.

(b) This proof mainly follows (Tveit, 2003). By Theorem 28.1 and Theorem 28.2 in (Cormen et al., 2009), let $I(n)$ be the complexity of inverting any $n \times n$ nonsingular matrix, and $M(n)$ be the complexity of multiplying two $n \times n$ matrices, then $I(n) = \Theta(M(n))$. That is, the complexity of matrix inversion is asymptotically both upper- and lower-bounded by the complexity of matrix multiplication.

Using the Coppersmith-Winograd algorithm (Coppersmith & Winograd, 1987), one of the fastest known algorithms for multiplying two $n \times n$ matrices, we have $M(n) = O(n^{2.376})$, which gives the upper bound $I(n) = O(n^{2.376})$.

Moreover, (Raz, 2002) proved that the complexity multiplying two $n \times n$ matrices cannot be better than $M(n) = \Omega(n^2 \log n)$. Thus $I(n) = \Omega(n^2 \log n)$.

(c) Following (b), we have the upper bound $M(n) = O(n^{2.376})$ and the lower bound $M(n) = \Omega(n^2 \log n)$.

$\square$

Corollary C.1 shows that it is possible to reduce the complexity of the Fast Algorithm to $O((2k + n^2)^{1.346} + n^{2.376})$ by choosing efficient algorithms for matrix multiplication and inversion; however, it is impossible to reduce it below $\Omega(n^2 \log n)$. It is easy to check that the same conclusion also applies to computing the closed-form solutions of EASE and EDLAE.

In the proof, the algorithms of Coppersmith-Winograd and Abboud et al. are used to show that this low complexity is theoretically achievable; however, they may not be practical to implement.

## D    DEQL with Low-Rank Constraint

This section discusses the closed-form solution of (7) under low-rank constraint of $W$. Given the rank $k$ $(k \leq n)$, we would like to solve

$$\underset{W}{\arg\min} \, l_{\mathcal{D}}(W) = \sum_{i=1}^{n} \mathbb{E}_{(X,Y) \sim \mathcal{D}^{(i)}} \left[ \|Y_{*i} - XW_{*i}\|_F^2 \right] \quad \text{s.t. } \text{rank}(W) \leq k \qquad (34)$$

**Theorem D.1.** *Suppose $\mathbb{E}_{(X,Y)\sim\mathcal{D}^{(i)}}[X^TX]$ is independent of $i$, and denote*

$$\Sigma_{xx} = \mathbb{E}_{(X,Y)\sim\mathcal{D}^{(i)}}[X^TX]$$
$$\Sigma_{xy} = \left[\mathbb{E}_{(X,Y)\sim\mathcal{D}^{(1)}}[X^TY_{*1}], \mathbb{E}_{(X,Y)\sim\mathcal{D}^{(2)}}[X^TY_{*2}], ..., \mathbb{E}_{(X,Y)\sim\mathcal{D}^{(n)}}[X^TY_{*n}]\right]$$

*If $\Sigma_{xx}$ is non-singular, then the closed-form solution of (34) is given by*

$$W^* = \Sigma_{xx}^{-1/2}\left[\Sigma_{xx}^{-1/2}\Sigma_{xy}\right]_k \tag{35}$$

*Here, let $\Sigma_{xx}^{-1/2}\Sigma_{xy} = U\begin{bmatrix}\sigma_1 & & & \\ & \sigma_2 & & \\ & & ... & \\ & & & \sigma_n\end{bmatrix}V^T$ be the singular value decomposition where $\sigma_1 \geq \sigma_2 ... \geq \sigma_n$, we denote $\left[\Sigma_{xx}^{-1/2}\Sigma_{xy}\right]_k = \sum_{i=1}^k \sigma_i U_{*i}V_{*i}^T$.*

*Proof*: Denote $y^{(i)} = \mathbb{E}_{(X,Y)\sim\mathcal{D}^{(i)}}[Y_{*i}^TY_{*i}]$. By (7),

$$l_{\mathcal{D}}(W) = \sum_{i=1}^n W_{*i}^T\Sigma_{xx}W_{*i} - 2W_{*i}^T(\Sigma_{xy})_{*i} + y^{(i)}$$

$$= \sum_{i=1}^n \left(\Sigma_{xx}^{1/2}W_{*i}\right)^T\Sigma_{xx}^{1/2}W_{*i} - 2\left(\Sigma_{xx}^{1/2}W_{*i}\right)^T\Sigma_{xx}^{-1/2}(\Sigma_{xy})_{*i} + y^{(i)}$$

$$= \sum_{i=1}^n \left\|\Sigma_{xx}^{1/2}W_{*i} - \Sigma_{xx}^{-1/2}(\Sigma_{xy})_{*i}\right\|_F^2 + y^{(i)} - (\Sigma_{xy})_{*i}^T\Sigma_{xx}^{-1}(\Sigma_{xy})_{*i}$$

$$= \left\|\Sigma_{xx}^{1/2}W - \Sigma_{xx}^{-1/2}\Sigma_{xy}\right\|_F^2 + \sum_{i=1}^n y^{(i)} - (\Sigma_{xy})_{*i}^T\Sigma_{xx}^{-1}(\Sigma_{xy})_{*i}$$

By Eckart–Young–Mirsky theorem, $\left[\Sigma_{xx}^{-1/2}\Sigma_{xy}\right]_k = \underset{\text{rank}(Q)\leq k}{\text{argmin}}\left\|Q - \Sigma_{xx}^{-1/2}\Sigma_{xy}\right\|_F^2$ for any $Q \in \mathbb{R}^{n\times n}$. Therefore, $\Sigma_{xx}^{1/2}W^* = \left[\Sigma_{xx}^{-1/2}\Sigma_{xy}\right]_k \implies W^* = \Sigma_{xx}^{-1/2}\left[\Sigma_{xx}^{-1/2}\Sigma_{xy}\right]_k$. $\square$

Note that the low-rank solution (35) is not applicable when $\Sigma_{xx} = \mathbb{E}_{(X,Y)\sim\mathcal{D}^{(i)}}[X^TX]$ depends on $i$: If $\Sigma_{xx}$ varies with $i$, then in the proof $\sum_{i=1}^n \left\|\Sigma_{xx}^{1/2}W_{*i} - \Sigma_{xx}^{-1/2}(\Sigma_{xy})_{*i}\right\|_F^2$ cannot be combined into $\left\|\Sigma_{xx}^{1/2}W - \Sigma_{xx}^{-1/2}\Sigma_{xy}\right\|_F^2$.

The low-rank solution (35) is applicable to EDLAE only when taking $a = b$: by (10), $\Sigma_{xx}$ is represented by $H^{(i)}$; by Lemma 3.2, if $a = b$, $H^{(i)}$ will have all diagonal entries being $(1-p)b^2$ and all off-diagonal entries being $(1-p)^2b^2$ for any $i$, thus being independent of $i$. However, when $a \neq b$, $H^{(i)}$ will depend on $i$, making (34) not applicable.

## E  RELATED WORKS

The evolution of collaborative filtering (CF) in recommendation systems has undergone several key paradigm shifts. In its early stages, neighborhood-based methods dominated the field, with influential works such as user-item KNN approaches (Hu et al., 2008) and sparse linear models (SLIM) (Ning & Karypis, 2011) setting the foundation. However, the Netflix Prize competition marked a turning point, accelerating the adoption of matrix factorization (MF) techniques, which offered improved scalability and latent feature learning (Koren et al., 2009). These models aim to solve a matrix completion problem, which has been extensively studied theoretically (Candès & Tao, 2010; Recht, 2011; Foygel et al., 2011; Shamir & Shalev-Shwartz, 2011).

The rise of deep learning (LeCun et al., 2015) further revolutionized the landscape, introducing more expressive neural architectures. Among these, graph-based models gained prominence, including Neural Collaborative Filtering (NCF) (He et al., 2017), which replaced traditional MF with

neural networks, and later refinements like Neural Graph Collaborative Filtering (NGCF) (Wang et al., 2019) and LightGCN (He et al., 2020), which explicitly leveraged graph structures for higher-order user-item relationship modeling. Simultaneously, industry-scale solutions emerged, blending memorization and generalization through hybrid architectures such as Wide & Deep (Cheng et al., 2016), DeepFM (Guo et al., 2017), and Deep & Cross Networks (DCN) (Wang et al., 2017), which automated feature interactions while maintaining interpretability.

Alongside the ongoing research that explores various models to enhance recommendation performance, the research community has gradually recognized the importance of gaining a deeper theoretical understanding of loss functions (Terven et al., 2025; Wu et al., 2024b). These theoretical investigations seek to reveal the fundamental principles and mathematical underpinnings that govern the behavior and optimization direction of recommendation systems, thereby advancing our overall understanding of how these systems function. BPR (Rendle et al., 2009). Early methods often adopted pointwise $L_2$ loss over observed ratings or implicit feedback (Hu et al., 2008), which is simple and analytically tractable. Later, pairwise ranking losses such as BPR (Rendle et al., 2009) became popular for top-K recommendation, optimizing relative preferences between positive and negative items. Softmax-based listwise losses, such as sampled softmax (Jannach et al., 2010) were introduced to better align with ranking metrics like NDCG. In recent years, contrastive learning frameworks (Zhou et al., 2021; Wang et al., 2022) have gained prominence as a powerful and effective approach, particularly in unsupervised recommendation scenarios. Notable studies such as (Li et al., 2023; Liu et al., 2021) have further showcased their effectiveness in this domain.

Notably, recent studies (Steck, 2019; 2020; Moon et al., 2023) have shown that well-tuned linear models can outperform deep nonlinear models (Liang et al., 2018) on sparse implicit-feedback data, challenging the assumption that greater model expressiveness necessarily leads to better performance. Moreover, these models are typically neighborhood-based and item-based, offering several advantages over deep learning approaches: more accurate recommendations based on a small number of high-confidence neighbors, substantially lower training cost, more interpretable recommendation mechanisms, and greater stability (Nikolakopoulos et al., 2021).

LAEs are one type of the linear recommender models. One of the earliest LAE model is SLIM (Ning & Karypis, 2011), which trains the loss $\|R - RW\|_F^2$ with $L_1$ and $L_2$ regularizers, together with zero-diagonal and non-negativity constraints on $W$. EASE (Steck, 2019) simplifies SLIM by retaining only the $L_2$ regularizer and the zero-diagonal constraint. EDLAE (Steck, 2020) instead employs dropout and emphasis as an alternative strategy to mitigate overfitting. ELSA (Vančura et al., 2022) construct the model $W$ with a zero diagonal by enforcing $W = AA^T - I$ for some matrix $A$ subject to $\|A_{i*}\|_2^2 = 1$ for all $i$. (Moon et al., 2023) shows that a strict zero-diagonal constraint does not always yield the best performance, and that replacing it with a diagonal bounded by a small norm during training can improve results.

Finally, an important research direction for LAE models is interpretability. LAE-based architectures provide a uniquely transparent mapping between input and output representations through a single linear operator $W$, where each element $W_{ij}$ quantifies how item $i$ contributes to predicting item $j$. A larger magnitude of $W_{ij}$, regardless of sign, indicates a stronger relatedness from $i$ to $j$ (Spišák et al., 2024). This white-box structure makes LAEs inherently interpretable compared with matrix factorization and deep neural recommenders, whose latent dimensions are unidentifiable or highly nonlinear. This interpretability direction aligns with broader developments in machine learning, where linear and sparse representations are increasingly used to reveal structure inside complex neural systems. In particular, Sparse Autoencoders (SAEs) trained on large language models have been shown to uncover highly interpretable and often monosemantic latent features Cunningham et al. (2023); Fel et al. (2025); Demircan et al. (2025). These findings extend earlier insights that deep activations can be linearly decomposed into disentangled semantic directions, a principle also supported by linear probes Alain & Bengio (2017). Complementary approaches such as LIME Ribeiro et al. (2016) further demonstrate how local linear surrogates can explain the predictions of arbitrary black-box models, reinforcing the idea that linearity provides a powerful lens for interpretability. Building on this insight, our proposed DEQL framework extends the explanatory power of LAEs, preserving their interpretability while enhancing expressive capacity.

# F  Supplemental Experiments

## F.1  Statistical Significance Test

Since the DEQL model is obtained via closed-form solution rather than gradient descent, each $b$ corresponds to a unique output model, thus there is no randomness in the training process. The only source of randomness arises from the train/validation/test data splitting. So it is sufficient to show that the performance improvements are unlikely affected by this type of randomness.

Table 2 reports results on a single dataset split. We fix the DEQL(L2) and DEQL(L2+zero-diag) models from Table 2, generate five random train/validation/test splits for each dataset, and evaluate each model on each split. The mean performance and standard deviations are reported in Table 4, and pairwise t-tests are presented in Table 5. These analyses consistently show that the performance improvements of DEQL(L2) and DEQL(L2+zero-diag) are stable and statistically significant.

Table 4: Mean performance ($\pm$ standard deviation) of DEQL(L2+zero-diag), DEQL(L2), and ED-LAE.

| Model | Games | | Beauty | | Gowalla | | ML20M | | Netflix | | MSD | |
|---|---|---|---|---|---|---|---|---|---|---|---|---|
| | R@20 | N@20 | R@20 | N@20 | R@20 | N@20 | R@20 | N@20 | R@20 | N@20 | R@20 | N@20 |
| EDLAE | $0.2674 \pm 0.0144$ | $0.1729 \pm 0.0030$ | $0.1526 \pm 0.0194$ | $0.0969 \pm 0.0164$ | $0.2266 \pm 0.0019$ | $0.2061 \pm 0.0027$ | $0.3971 \pm 0.0031$ | $0.3482 \pm 0.0026$ | $0.3657 \pm 0.0012$ | $0.3434 \pm 0.0006$ | $0.3317 \pm 0.0006$ | $0.3233 \pm 0.0007$ |
| DEQL(L2+diag) | $0.2669 \pm 0.0188$ | $0.1737 \pm 0.0027$ | $0.1517 \pm 0.0225$ | $0.1001 \pm 0.0148$ | $0.2270 \pm 0.0026$ | $0.2071 \pm 0.0031$ | $0.3972 \pm 0.0036$ | $0.3482 \pm 0.0027$ | $0.3657 \pm 0.0013$ | $0.3433 \pm 0.0006$ | $\mathbf{0.3344} \pm 0.0005$ | $0.3247 \pm 0.0015$ |
| DEQL(L2) | $\mathbf{0.2745} \pm 0.0153$ | $\mathbf{0.1738} \pm 0.0067$ | $\mathbf{0.1567} \pm 0.0208$ | $\mathbf{0.1023} \pm 0.0164$ | $\mathbf{0.2279} \pm 0.0023$ | $\mathbf{0.2076} \pm 0.0029$ | $\mathbf{0.3974} \pm 0.0032$ | $\mathbf{0.3484} \pm 0.0026$ | $\mathbf{0.3660} \pm 0.0012$ | $\mathbf{0.3437} \pm 0.0006$ | $0.3333 \pm 0.0005$ | $\mathbf{0.3251} \pm 0.0007$ |

Table 5: Pairwise t-tests results at significance level of $\alpha = 0.05$. The null hypothesis $A \leq B$ means that method $A$ performs no better than method $B$. Each cell reports the decision to reject (Rej) or accept (Acc) the null hypothesis and the corresponding p-value. If p-value $< \alpha$, the null hypothesis is rejected (i.e., the evidence supports that $A$ performs better than $B$).

| Null Hypothesis | Games | | Beauty | | Gowalla | | ML20M | | Netflix | | MSD | |
|---|---|---|---|---|---|---|---|---|---|---|---|---|
| | R@20 | N@20 | R@20 | N@20 | R@20 | N@20 | R@20 | N@20 | R@20 | N@20 | R@20 | N@20 |
| DEQL(L2) $\leq$ DEQL(L2+diag) | Rej (p=0.0411) | Acc (p=0.4884) | Acc (p=0.1486) | Rej (p=0.0353) | Rej (p=0.0134) | Rej (p=0.0164) | Acc (p=0.1924) | Acc (p=0.0523) | Rej (p=0.0171) | Rej (p=0.0001) | Acc (p=0.9938) | Acc (p=0.2165) |
| DEQL(L2) $\leq$ EDLAE | Rej (p=0.0137) | Acc (p=0.4008) | Acc (p=0.0729) | Rej (p=0.0013) | Rej (p=0.0083) | Rej (p=0.0001) | Rej (p=0.0033) | Rej (p=0.0400) | Rej (p=0.0401) | Rej (p=0.0001) | Rej (p=0.0000) | Rej (p=0.0000) |

## F.2  Learned Diagonal Values in DEQL(L2)

We visualize the distributions of the diagonal values learned by DEQL(L2) in Figure 3. The results show that the diagonal entries of the optimal DEQL(L2) models are typically small, with the vast majority remaining close to zero and sharp modes generally falling within the range of 0.01–0.10 across datasets.

This pattern indicates that even without strict $\mathrm{diag}(W) = 0$ constraint, the $L_2$ regularizer is still able to effectively suppress the diagonal terms, preventing them from growing into large. This is consistent with the findings of (Moon et al., 2023), which demonstrate that relaxing the zero-diagonal constraint to allow small diagonal values can enhance performance.

Moreover, the optimal models on Yelp2018 and AmazonBook exhibit relatively larger diagonal values than those on other datasets, suggesting that the model places greater emphasis on the self-reconstruction of remained items. This observation may help explain the counterintuitive results in Figure 1, where the $b/a$ ratio corresponding to peak performance on these two datasets exceeds 1.

## F.3  Time and Memory Cost

In this section, we compare the time and memory costs of DEQL with existing LAE-based and deep learning–based models. All models are evaluated on the Yelp2018 dataset. The LAE-based models are trained on CPU and solved via closed-form solutions, whereas the deep learning–based models are trained on GPU via gradient descent.

The results are presented in Table 6, which show that LAE-based methods (including DEQL) exhibit higher memory consumption but lower training time than deep learning-based methods, *even without using a GPU*. Note that LAE-based models and deep learning models follow distinct computational paradigms. Deep learning-based models train via batch gradient descent, loading only small batches

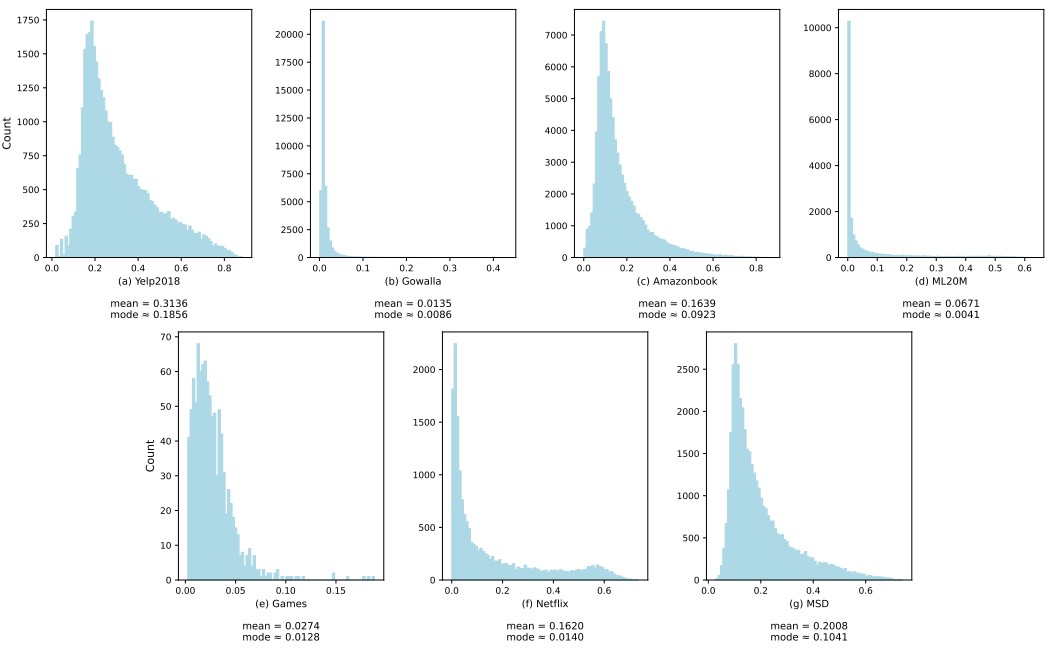

Figure 3: Diagonal Value Distribution Across different datasets

of data into GPU memory at each step, which keeps memory usage around 1GB but requires many iterations, leading to longer training time. In contrast, LAE-based methods load the entire matrix into CPU memory to compute its inverse as part of a closed-form solution, which may require 80GB for datasets like Yelp2018. This space-time trade-off enables the entire training process to complete much faster.

Unlike GPU-intensive GNN-based models such as LightGCN or SSM (GNN), DEQL models are well suited for memory-limited or CPU-only environments. Modern CPU servers typically provide 500 GB - 1 TB of RAM, suggesting that memory limitations are unlikely to pose practical concerns. Moreover, as discussed in Appendix G.3, moderately sized DEQL models (as well as other LAE models) can be readily adapted to a GPU environment for further acceleration.

Table 6: Approximate Training time and memory usage on Yelp2018. Deep based models mainly consume GPU memory, while others rely on CPU memory.

| Model | Time (min) | Memory (GB) | Memory Type |
|---|---|---|---|
| LightGCN | 30 | 1 | GPU |
| SimpleX (GNN) | 130 | 1 | GPU |
| SSM (MF) | 13 | 1 | GPU |
| SSM (GNN) | 13 | 1 | GPU |
| DLAE | 2 | 70 | CPU |
| EASE | 2 | 60 | CPU |
| EDLAE | 4 | 70 | CPU |
| DEQL | 6 | 80 | CPU |
| DEQL (L2+zero-diag) | 8 | 80 | CPU |
| DEQL (L2) | 8 | 80 | CPU |

# G DISCUSSIONS

## G.1 LIMITATIONS

One limitation of our work is that DEQL is currently used only as an optimization tool, providing closed-form solutions under given hyperparameters. However, it does not offer theoretical guidance on which hyperparameter choices lead to improved testing performance. As a result, hyperparameter selection still relies on empirical tuning, and its theoretical understanding remains underexplored. Our experiments show that datasets with a larger item-user ratio tend to perform well under large $b$ (Section 5.3).

## G.2 CONNECTION BETWEEN EDLAE OBJECTIVE AND MEAN SQUARED ERROR

In evaluation, the interaction matrix $R$ typically adopt a *hold-out* split (Section 7.4.2, (Aggarwal, 2016)): Let $\Delta \in \{0,1\}^{m \times n}$ be a mask matrix, and $\mathcal{B}_R$ denote a multivariate Bernoulli distribution over $\Delta$ and conditioned on $R$, such that each $\Delta_{ij}$ independently satisfies $P(\Delta_{ij} = 0|R_{ij} = 1) = p$, $P(\Delta_{ij} = 1|R_{ij} = 1) = 1 - p$ and $P(\Delta_{ij} = 0|R_{ij} = 0) = 1$ for a given $p \in (0,1)$. Then $\Delta \odot R$ represents the random hold out operation on $R$: for any $R_{ij} = 1$, if $\Delta_{ij} = 0$, then $R_{ij}$ is held out. Given any fixed model $W$ and dataset $R$, the *Mean Squared Error (MSE)*, a commonly used evaluation metric (Section 7.5.1, Aggarwal (2016)), can be viewed as a function of $\Delta$:

$$\text{MSE}_{W,R}(\Delta) = \frac{1}{\|\Delta\|_F^2} \|(\mathbf{1} - \Delta) \odot (R - (\Delta \odot R)W)\|_F^2 \tag{36}$$

where $\mathbf{1} \in \{1\}^{m \times n}$ denotes the all-one matrix and $\|\Delta\|_F^2$ equals the number of non-zeros in $\Delta$. When $\text{MSE}_{W,R}(\Delta)$ is used as the test error, we draw a realization $\Delta \sim \mathcal{B}_R$ and evaluate it.

The objective function of EDLAE can be interpreted as an evolution of (36) obtained by

1. Removing the scalar $\frac{1}{\|\Delta\|_F^2}$.

2. Replacing $\mathbf{1} - \Delta$ with an *emphasis matrix* $A \in \{a, b\}^{m \times n}$. If $(\mathbf{1} - \Delta)_{ij} = 1$, then $A_{ij} = a$; if $(\mathbf{1} - \Delta)_{ij} = 0$, then $A_{ij} = b$, where $a \geq b$. This is equivalent to placing greater emphasis on held-out (dropped) entries.

3. Relaxing $\mathcal{B}_R$ to $\mathcal{B}$ and taking the expectation over $\mathcal{B}$, where $\Delta \sim \mathcal{B}$ indicates that the entries $\Delta_{ij}$ are i.i.d. Bernoulli random variables with $P(\Delta_{ij} = 0) = p$ and $P(\Delta_{ij} = 1) = 1 - p$.

Consequently, the objective can be written as

$$f(W) = \mathbb{E}_{\Delta \sim \mathcal{B}} \left[ \|A \odot (R - (\Delta \odot R)W)\|_F^2 \right] \tag{37}$$

This observation shows that the EDLAE training objective aligns with the MSE used for evaluation. Moreover, previous empirical studies have shown that models with lower MSE are typically associated with better ranking performance, including a higher probability of relevant items appearing in top-ranked positions (Koren, 2008). In addition, (Guo et al., 2025) derive a generalization bound based on a relaxed MSE and show that a tighter bound is often correlated with improved top-K ranking metrics such as Recall and NDCG.

Hence, one might intuitively expect that, given the alignment between EDLAE and MSE, minimizing the EDLAE objective would improve Recall and NDCG. However, our experimental results in Section 5.3 show that this intuition does not always hold, as models in the $b > a$ case, which no longer strictly align with MSE, can in some cases achieve superior Recall and NDCG performance.

## G.3 PRACTICAL SCALABILITY OF DEQL

The LAE models produced by DEQL or other methods are typically represented by an $n \times n$ matrix. Here, *practical scalability* concerns how to handle, for large $n$, both the *inefficiency* of the Fast Algorithm (Section 4) used to compute the model and the associated *memory prohibitivity* issues. Since computation is often performed on GPUs – which generally have much smaller memory capacity than CPU RAM – we discuss the scalability challenge at two levels of memory prohibitivity:

**1. The $n \times n$ matrix fits in CPU RAM but cannot fit in GPU memory**: At this level, we do not need to impose rank constraints, and the model can remain full-rank. The issue is computational: the Fast Algorithm depends on matrix multiplication and inversion, which are too slow on CPU. Using the GPU would accelerate these operations, but matrices such as $R$ or $R^T R$ are too large to load into GPU for a single-pass computation. Instead, we can use block matrix multiplication and block matrix inversion. These methods partition $R$ or $R^T R$ into blocks that fit in GPU memory and can be processed sequentially.

**2. The $n \times n$ matrix cannot fit in CPU RAM**: In this case, a low-rank LAE can be used instead of a full-rank model. Importantly, the number of parameters in an LAE can be freely scaled. For example, ELSA (Vančura et al., 2022) represents the model as $A^T A - I \in \mathbb{R}^{n \times n}$ where $A \in \mathbb{R}^{l \times n}$. Although the final model is still an $n \times n$ matrix, the number of parameters depends on the size of $A$, which can be flexibly controlled through $l$. Choosing $l < n$ produces a low-rank LAE. Moreover, as shown in Appendix D, a low-rank DEQL closed-form solution can also be obtained in the restricted case $a = b$.

Additionally, to accelerate computation, one may replace the standard $O(n^3)$ matrix multiplication algorithm with Strassen's algorithm, which runs in $O(n^{2.81})$. Unlike the Coppersmith–Winograd algorithm discussed in Appendix C, Strassen's algorithm remains practical to implement.

### G.4   ADVANTAGES OF CLOSED-FORM SOLUTIONS

DEQL provides a framework for computing and analyzing closed-form solutions. A closed-form solution guarantees the global optimum of the training objective, but it does *not necessarily* yield the best test performance, as it may overfit the training data. In contrast, gradient-based optimization can rely on *early stopping* as a regularization technique (Yao et al., 2007) to mitigate overfitting, in which case the resulting model does *not* minimize the training objective.

However, the closed-form solution has a distinct advantage in hyperparameter tuning. When hyperparameters are fixed, gradient-based training typically produces non-deterministic models due to the reliance on early stopping, which introduces randomness in addition to the randomness from data splitting. This compounded uncertainty makes it harder to assess whether a particular hyperparameter setting truly leads to good generalization.

By contrast, the closed-form solution is fully deterministic for any fixed hyperparameters, so the only source of randomness stems from the data itself. This substantially reduces uncertainty in model evaluation and makes hyperparameter tuning more reliable. Consequently, closed-form solutions enable cleaner hyperparameter selection and facilitate reproducibility.

### G.5   LAEs VERSUS DEEP MODELS

Deep neural networks have become popular in industrial recommender systems due to their flexibility: their depth and width are not constrained by the dataset dimension $n$. In contrast, LAEs are typically represented by an $n \times n$ matrix, inherently tied to $n$. Although their parameter count can be scaled while preserving the $n \times n$ size – such as the ELSA model described in Appendix G.3 – these models remain comparatively shallow. As a result, deep models generally benefit from more flexible parameter scaling and architectural design, which often yields stronger representation power.

However, this does not imply that LAEs always underperform deep networks. Empirical results show that LAE models outperform deep neural networks on sparse datasets (Steck, 2019; 2020; Moon et al., 2023). Such sparsity is common in recommender datasets (Table 1), since most users and items are associated with only a few ratings. LAE models have been shown to effectively capture item-item correlations using a small number of high-confidence neighbors, thereby achieving high accuracy (Nikolakopoulos et al., 2021). Many small- to medium-scale e-commerce platforms have abundant interaction logs but limited or no side features – a scenario that characterizes a substantial portion of industrial deployments. In such environments, LAE-style models and matrix factorization remain both effective and operationally attractive (Dacrema et al., 2019).

Moreover, unlike deep learning based models, LAEs are typically less affected by the constant addition of users, items and ratings, which are typically observed in large commercial applications.

For example, once item similarities have been computed, an item-based system can readily make recommendations to new users, without having to re-train the system (Nikolakopoulos et al., 2021).

Finally, linear models are typically easier to analyze than deep models due to their simplicity and structural stability, which makes them widely used in interpretable AI (Ribeiro et al., 2016). In particular, LAEs have been regarded as interpretable recommender systems. (Spišák et al., 2024) show that the magnitude of each $W_{ij}$, regardless of sign, represents the relatedness from item $i$ to item $j$; hence, item relationships as a directed graph, noting that the weight from $i$ to $j$ is not necessarily equal to the weight from $j$ to $i$. These interpretable relationships are a key reason why LAE-style models remain prominent in production systems, compared with deep learning-based models whose learned weights are often difficult to explain. Although DEQL was originally developed to enable closed-form theoretical analysis for LAE models, the resulting LAE solutions can also be naturally leveraged for interpretability at the application level.

### G.6 BROADER SCOPE OF LAEs AND DEQL

Below, we highlight several non-Recommender Systems application domains in which LAE-style models and DEQL may be particularly useful. These domains overlap with LAE-related topics such as matrix completion, linear regression, and autoencoders:

- **Survey & psychometric modeling; genomics & biomedical panels**: In survey research, user $\times$ item response matrices exhibit substantial missingness and heterogeneous exposure, a setting long modeled using linear or matrix-completion methods (Mazumder et al., 2010; Yoon et al., 2018). Similarly, genomics, metabolomics, and biomedical panels routinely rely on linear or low-rank imputation methods for patient $\times$ gene and panel-level data, including mass-spectrometry-based metabolomics (Stekhoven & Bühlmann, 2012; Wei et al., 2018).
- **Distributed Sensing & Sensor Network**: In distributed sensing applications, sensortime matrices frequently contain missing measurements due to intermittent connectivity, power constraints, or sensor failures. Linear reconstruction and imputation methods continue to be widely used in these resource-constrained environments, where computational simplicity and interpretability are critical (Rivera-Muñoz et al., 2022).
- **LLM Interpretability**: Sparse Autoencoders (SAEs) have emerged as a key tool for interpreting LLMs (Fel et al., 2025; Cunningham et al., 2023). A typical SAE can be formulated as follows (Cunningham et al., 2023). Let $x$ be the input feature vector, $\hat{x}$ be the reconstructed feature vector, $c$ be the latent vector, and $W$ be the parameter matrix. The encoder is defined as $c = \phi(Wx + b)$ for some non-linear activation $\phi$, and the decoder is given by $\hat{x} = W^T c$. The model learns a $W$ such that $x$ and $\hat{x}$ are close under a reconstruction loss, while $c$ is regularized to be sparse. The sparsity of $c$ facilitates the identification of important features, thereby improving interpretability. This is similar to a LAE model, where $W$ can be factorized into two matrices $W = A^T B$, with $c = Bx$ be the encoder and $\hat{x} = A^T c$ be the decoder. In this formulation, the latent vector $c$ can be regularized to be sparse if desired.

Finally, we note a conceptual connection to LLM preference learning (which is related to recommendation, though not necessarily restricted to linear models). While methods such as DPO differ substantially in their mathematical formulation, both LLM-based content generation and recommendation involve selecting or ranking items from large discrete spaces based on preference signals (Rafailov et al., 2023; Rendle et al., 2009).

Although direct application would require substantial methodological development beyond DEQL's current scope, these connections suggest interesting directions for future research.

### LLM USAGE STATEMENT

We use ChatGPT solely for polishing writing at the sentence and paragraph level. The content and contributions of this paper were created by the authors. All text refined with ChatGPT has been carefully checked to avoid factual errors.

