# OpenReview forum: "Generalizing Linear Autoencoder Recommenders with Decoupled Expected Quadratic Loss"
_ICLR.cc/2026/Conference — ICLR 2026 Poster_

### Official Review · Reviewer_9r8b · 2025-10-24

**Soundness:** 4
**Presentation:** 4
**Contribution:** 3
**Rating:** 6
**Confidence:** 3

**Summary:**

This paper addresses a key limitation in a state-of-the-art linear recommendation model, the 'Emphasized Denoising Linear Autoencoder' (EDLAE). The original EDLAE work only provided a closed-form solution for the specific hyperparameter case of b=0, which limited the model's capacity.

The authors introduce the 'Decoupled Expected Quadratic Loss' (DEQL), a generalized objective function that reformulates the EDLAE problem from a statistical perspective. The main contributions of this work are:

1. **Theoretical Generalization:** Using the DEQL framework, the authors derive a general closed-form solution (minimizer) for the expected quadratic loss.
2. **Expanded Solution Space:** They apply this general solution back to the EDLAE objective, successfully deriving closed-form solutions for the previously unexplored hyperparameter regime of b>0.
3. **Efficient Algorithm:** Recognizing that a naive computation of these new b>0 solutions is computationally prohibitive ($O(n^4)$), the authors propose a novel and efficient algorithm based on the Miller matrix inversion theorem.
4. **Complexity Reduction:** This algorithm reduces the complexity to $O(n^3)$, making it practical and comparable to existing methods like EASE and the b=0 EDLAE baseline.
5. **Empirical Validation:** Extensive experiments on six benchmark datasets show that the newly derived solutions for b>0 (especially when combined with $L_2$ regularization, termed DEQL(L2)) consistently and significantly outperform the original b=0 EDLAE baseline.
6. Novel Insight: The paper also provides an interesting insight that the explicit zero-diagonal constraint is not strictly necessary for strong performance when b>0.

**Strengths:**

- **Originality and Theoretical Quality:** The generalization of the EDLAE objective into the Decoupled Expected Quadratic Loss (DEQL) is an elegant and novel theoretical contribution. It provides a clearer and more principled statistical framework for understanding this class of models. Deriving the closed-form solution for the b>0 case is a non-trivial and significant extension of prior work.
- **Practical Algorithmic Contribution:** A major strength is that the authors did not stop at the computationally intractable $O(n^4)$ solution. The development of the $O(n^3)$ algorithm using the Miller matrix inversion theorem is a crucial contribution that makes the entire theoretical framework practical and usable. This new algorithm maintains the same complexity as the original EASE and EDLAE (b=0) models.
- **Clarity and Presentation:** The paper is extremely well-written and easy to follow. The motivation is clear, and the logical flow—from the limitation of EDLAE, to the general DEQL framework, to the specific b>0 solution, and finally to the efficient algorithm—is impeccable. The mathematical derivations are clean and well-explained.
- **Significance and Empirical Results:** The empirical validation is thorough and convincing. The proposed b>0 solution (specifically DEQL(L2)) demonstrates consistent and sometimes substantial performance gains over the strong EDLAE baseline across all six datasets. The paper also reinforces the value of simple linear models, showing they remain competitive and, in some cases, superior to complex deep learning-based models while offering significantly faster (CPU-based) training.
- Novel Insight: The finding (RQ2) that the zero-diagonal constraint—essential in models like EASE and EDLAE—is not strictly necessary when b>0 and $L_2$ regularization is applied is an important insight for future model development in this area.

**Weaknesses:**

1. **Lack of Theoretical Guidance on Hyperparameters:** The DEQL framework successfully expands the solution space to b>0, but it does not provide theoretical guidance on *how* to select the optimal hyperparameters (e.g., the ratio b/a, p, or the $L_2$ coefficient $\lambda$). Navigation of this newly expanded solution space still relies on empirical grid search, which can be costly. The paper proves *that* b>0 solutions are better, but not *which* b>0 solution is optimal a priori.
2. **Computational Bottleneck:** While the $O(n^3)$ algorithm is a significant achievement, it still relies on the inversion of an $n \times n$ matrix ($H_0$ or $H_0 + \lambda I$). For datasets with a very large number of items (n), this $O(n^3)$ step remains a significant bottleneck, limiting its scalability into the high tens of thousands. This is an inherent limitation of this class of closed-form LAE models, but it remains a practical weakness.
3. **'Pure' DEQL Performance:** The core theoretical contribution is the 'pure' DEQL solution (Eq. 9). However, the experimental results in Table 1 clearly show that this pure DEQL model performs poorly and is often worse than the EASE/EDLAE baselines. The strong performance gains are only realized *after* adding $L_2$ regularization (DEQL(L2)) or both $L_2$ and the zero-diag constraint. This suggests the primary practical contribution is not DEQL *per se*, but *regularized* DEQL. This is not a major flaw, but the paper's framing could be slightly more precise to reflect that the b>0 solution *requires* regularization (like $L_2$) to be effective, though this is standard practice anyway.
4. The work's key contribution is performance uplift. However, the experimental gains shown in Table 1 are not statistically significant. For example, on ML-20M, Recall@20 improves from 0.3925 to 0.3934, and NDCG@20 from 0.3421 to 0.3426. On Netflix, the performance is identical. On MSD, the b=0 EDLAE baseline actually outperforms the proposed DEQL(L2+zero-diag). These minor and inconsistent gains hardly justify adopting a new solution, let alone the extra hyperparameter tuning for b.
5. Concurrent research is working to break this cubic complexity bottleneck, e.g., via scalable iterative methods, graph-based approximations, or randomized linear algebra. The paper's contribution lacks some theoretical or experimental comparisons in terms of effectiveness and complexity with these works.
6. The paper adds a new hyperparameter (b) to an already complex tuning space (dropout p, $L_2$ $\lambda$). This runs counter to the broader research trend of seeking simplification for robust performance. Other recent work suggests that the dropout and $L_2$ terms already provide sufficient regularization, making an explicit b parameter slightly redundant.

**Questions:**

1. **Sensitivity to b:** Related to Weakness 1, how sensitive is the model's performance to the choice of b (or the b/a ratio)? The grid search ranged from [0.1, 2.0]. Could you provide a brief analysis (perhaps with a plot) on the validation set showing how Recall@20 varies as b changes? Is there a clear 'sweet spot,' or is performance relatively flat across a wide range of b>0?
2. **Statistical Significance:** The experimental gains in Table 1 (ML-20M, Netflix) appear to be well within the margin of error. Can the authors please provide statistical significance tests (e.g., paired t-tests) across multiple runs to demonstrate that the b>0 solutions are *truly* superior to the b=0 baseline? As it stands, the evidence suggests the gains are just noise from tuning an extra parameter.
3. **Learned Diagonal Values:** You argue convincingly in RQ2 that the zero-diagonal constraint is not strictly necessary when b>0 and $L_2$ regularization is used (DEQL(L2)). This is a very interesting finding. In the DEQL(L2) model, what do the diagonal entries of the learned $W^*$ matrix look like? Are they small values close to zero (suggesting the $L_2$ norm effectively pushes them down), or do they take on meaningful non-zero values? This could offer insight into *why* the hard constraint is no longer needed.
4. **Ablation of b vs. $\lambda$:** The 'naive' DEQL (Eq. 9) performs poorly, and strong results are only obtained with DEQL(L2). This suggests the $L_2$ regularization is doing the heavy lifting, not the b>0 formulation. Could the b=0 EDLAE baseline achieve the same performance as DEQL(L2) if its $L_2$ hyperparameter ($\lambda$) was simply tuned more aggressively over a wider range?
5. **Scalability and Approximation:** The $O(n^3)$ inversion of $H_0$ is the main bottleneck. Have the authors considered or can they comment on using approximation techniques for $(H_0 + \lambda I)^{-1}$? For example, would an iterative solver (e.g., Conjugate Gradient) or a low-rank approximation of $H_0$ be a viable path for scaling this method to n >> 50,000 items? This would break the pure 'closed-form' nature but might offer a practical path to larger datasets.
6. Generality of the Algorithm: The efficient $O(n^3)$ algorithm relies on $H^{(i)}$ being a low-rank (specifically, rank-2) update to a common matrix $H_0$. This structure arises from the specific dropout/emphasis scheme of EDLAE. Is this low-rank structure a fundamental property of DEQL, or just a property of this specific instantiation? In other words, could other formulations of DEQL (i.e., different choices for $\mathcal{D}^{(i)}$ in Def 3.1) lead to $H^{(i)}$ matrices that do not have this structure, making the Miller theorem speedup inapplicable?

---

> ### Author Response · Authors · 2025-11-21
>
> Thank you for your detailed and constructive comments. We hope our responses address your concerns, and we are happy to clarify further if needed.
>
> > W1/Q1: The paper proves that $b>0$ solutions are better, but not which $b>0$ solution is optimal a priori. ... How sensitive is the model's performance to the choice of b (or the b/a ratio)?
>
> We thank the reviewer for this thoughtful question.
>
> In **Section 5.4**, we have added an extended analysis showing how the ratio $b/a$ influences model performance. By varying $b/a$ in $[0, 2.0]$, we observe that the “sweet spot’’ typically occurs at $b > 0$ rather than $b = 0$, confirming that DEQL meaningfully extends the solution space beyond EDLAE. Interestingly, in some datasets the optimal region occurs at $b/a > 1$ (i.e., $b > a$). Although this region is counter-intuitive under the original EDLAE interpretation, it still yields better performance, indicating that emphasizing dropped entries is not always beneficial.
>
> Moreover, our analysis suggests that the user-item cardinality ratio (\#items/\#users) can help guide the selection of $b$. For datasets with many items but relatively few users (e.g., AmazonBook, Yelp2018), models with higher $b/a$ values perform better. We hypothesize that in these settings, increasing $b$ reduces reliance on weak cross-item correlations while strengthening self-reconstruction signals.
>
> > W2/W5/Q5: For datasets with a very large number of items $n$, this $O(n^3)$ step remains a significant bottleneck. ... Have the authors considered or can they comment on using approximation techniques?
>
> Thanks for raising the concerns about the $O(n^3)$ complexity of our algorithm.
>
> We provide a refined complexity analysis in **Section 4** and **Appendix C**, showing that if using more advanced algorithms for matrix multiplication and inversion, this complexity can be reduced to $O(n^{2.376})$, but cannot be improved beyond $\Omega (n^2\log n)$.
>
> Importantly, our analysis is still based on closed-form solution and does not involve any approximation.
>
> > W3/Q4: The strong performance gains are only realized after adding $L_2$ regularization (DEQL(L2)) or both $L_2$ and the zero-diag constraint.
>
> We clarify that EASE, DLAE and EDLAE all inherently include $L_2$ regularization controlled by $\lambda$. For EASE, this is shown in Eq (2) of [1]. For EDLAE, the caption of Table 1 in [2] explicitly states: "Note that $\tilde{\Lambda} = \Lambda + \lambda I$, ..., and the (scalar) L2-regularization parameter $\lambda$", indicating that an additional $L_2$ regularizer is used in their experiments. Since $L_2$ regularization is known to improve performance, and because EASE, DLAE and EDLAE all use $L_2$ regularization in our experiments, DEQL should also include a $L_2$ regularization to ensure a fair comparison.
>
> The $b=0$ EDLAE baseline we report already includes this $L_2$ regularization, and its results are shown in **Tables 1 and 2**.
>
> > W4/Q2: Can the authors please provide statistical significance tests (e.g., paired t-tests) across multiple runs to demonstrate that the $b>0$ solutions are truly superior to the $b=0$ baseline?
>
> Thanks for your suggestion! We have added the statistical significance tests in **Appendix F.1**. Note that for a fixed $b$, the closed-form solution is not random, so the only source of randomness is dataset splitting. We evaluate each model on five random splits of each dataset, compute standard deviations of R@20 and N@20, and perform paired t-tests. The results indicate that DEQL(L2)’s $b > 0$ solution reliably outperforms the EDLAE $b = 0$ baseline.
>
> > Q3: In the DEQL(L2) model, what do the diagonal entries of the learned matrix look like? Are they small values close to zero (suggesting the norm effectively pushes them down), or do they take on meaningful non-zero values?
>
> Thanks for your interests in our finding! To better illustrate the diagonal elements, we have added a visualization of their distribution in **Appendix F.2**. The results show that most diagonal values are close to 0, indicating that relaxing the zero-diagonal constraint to allow small values can improve performance.
>
> > W6: Other recent work suggests that the dropout and terms already provide sufficient regularization, making an explicit b parameter slightly redundant.
>
> In EDLAE, Steck showed that dropout induces an $L_2$ regularizer parameterized by the diagonal matrix $\Lambda$, (Eq (3) of [2]). However, in the experiment (caption of Table 1 in [2]), Steck additionally introduced another $L_2$ regularizer with parameter $\lambda$, so the final EDLAE uses two $L_2$ regularizers, $\tilde{\Lambda} = \Lambda + \lambda I$. This indicates that the two regularizers are not redundant; rather, using them together can yield better performance than relying on either one alone.
>
> **References:**
>
> [1] Harald Steck. Embarrassingly shallow autoencoders for sparse data. WWW, 2019.
>
> [2] Harald Steck. Autoencoders that don’t overfit towards the identity. NeurlIPS, 2020.

---

### Official Review · Reviewer_bLSH · 2025-10-30

**Soundness:** 3
**Presentation:** 3
**Contribution:** 2
**Rating:** 6
**Confidence:** 3

**Summary:**

This manuscript proposes a generalization of EDLAE named Decoupled Expected Quadratic Loss (DEQL), which extends the theoretical formulation and provides closed-form solutions for a wider range of hyperparameters range b>0. Moreover, this manuscript propose an efficient algorithm based on Miller’s matrix inverse theorem to to reduce the computational complexity for the b > 0 case. Extensive experiments demonstrate the advantages of DEQL on multiple benchmark datasets.

**Strengths:**

1. The paper clearly identifies a key limitation of the existing EDLAE model, the lack of analytical solutions for the case b>0. By introducing the DEQL formulation, the authors successfully generalize EDLAE and provide new theoretical insights into the design of expected quadratic loss functions. The proposed formulation is both mathematically and conceptually well-grounded.
2. The authors provide detailed derivations, lemmas, and proofs to support the theoretical claims. The transition from the expected quadratic loss formulation to the closed-form solution is rigorous and contributes to a deeper understanding of linear autoencoder objectives.
3. The introduction of the Miller-based algorithm significantly improves computational efficiency, reducing complexity from O(n4) to O(n3). This makes DEQL practical for large-scale recommendation datasets.
4. The datasets used are publicly available, and the implementation details are well described. The code link is also provided, which enhances transparency and reproducibility.

**Weaknesses:**

1. The entire paper contains no figures or diagrams to illustrate the overall framework, problem formulation, or intuition behind DEQL. The heavy use of equations makes it difficult for readers to intuitively understand the workflow and conceptual contribution. Adding schematic diagrams of the model or algorithm would greatly improve readability.
2. As the proposed method is a linear autoencoder-based (LAE-based) method, the most recent baseline considered in the paper is EDLAE (2020). Including more up-to-date LAE-based models for comparison would make the experimental evaluation more convincing and better demonstrate the advantages of the proposed DEQL framework..
3. The paper mentions in the experimental section that only the ratio b/a affects the closed-form solution. Therefore, the authors fix 𝑎 a and perform a grid search over b. Since the paper emphasizes that b≥0 enables exploration of a larger solution space, it would be important to further analyze how different values of b influence the experimental results. A detailed discussion or sensitivity analysis on b would strengthen the empirical section and clarify its effect on model performance.

**Questions:**

1. The entire paper contains no figures or diagrams to illustrate the overall framework, problem formulation, or intuition behind DEQL. The heavy use of equations makes it difficult for readers to intuitively understand the workflow and conceptual contribution. Adding schematic diagrams of the model or algorithm would greatly improve readability.
2. As the proposed method is a linear autoencoder-based (LAE-based) method, the most recent baseline considered in the paper is EDLAE (2020). Including more up-to-date LAE-based models for comparison would make the experimental evaluation more convincing and better demonstrate the advantages of the proposed DEQL framework..
3. The paper mentions in the experimental section that only the ratio b/a affects the closed-form solution. Therefore, the authors fix 𝑎 a and perform a grid search over b. Since the paper emphasizes that b≥0 enables exploration of a larger solution space, it would be important to further analyze how different values of b influence the experimental results. A detailed discussion or sensitivity analysis on b would strengthen the empirical section and clarify its effect on model performance.

---

> ### Author Response · Authors · 2025-11-21
>
> Thank you for your helpful and constructive feedbacks. We hope our responses address your concerns, and we are happy to clarify further if needed.
>
> > W1/Q1: The entire paper contains no figures or diagrams to illustrate the overall framework, problem formulation, or intuition behind DEQL.
>
> Thank you for pointing out this issue. We have added a figure illustrating the DEQL framework in Appendix A. Please let us know if further clarification is needed.
>
> > W2/Q2: As the proposed method is a linear autoencoder-based (LAE-based) method, the most recent baseline considered in the paper is EDLAE (2020). Including more up-to-date LAE-based models for comparison would make the experimental evaluation more convincing and better demonstrate the advantages of the proposed DEQL framework.
>
> Thanks for your comments. We have added the recent LAE model ELSA (Vancura et al., 2022) to the comparisons in **Tables 1 and 2**. The results show that DEQL(L2) and DEQL(L2+zero-diag) outperforms it.
>
> > W3/Q3: Since the paper emphasizes that $b\ge0$ enables exploration of a larger solution space, it would be important to further analyze how different values of b influence the experimental results.
>
> Thank you for raising this important concern.
>
> In **Section 5.4**, we have added experiments analyzing how the ratio $b/a$ influences model performance. By varying $b/a$ in $[0, 2.0]$, we observe that on certain datasets the “sweet spot’’ occurs at $b > 0$ rather than $b = 0$. Surprisingly, in some cases it even occurs at $b/a > 1$ (i.e., $b > a$). Although such settings are not meaningful in EDLAE (as they emphasize non-dropped entries more than dropped ones), they still yield better performance on some datasets. This suggests that emphasizing dropped entries does not universally improve performance and may sometimes hurt it.
>
> Moreover, our analysis suggests that the user-item cardinality ratio (\#items/\#users) can help guide the selection of $b$. For datasets with many items but relatively few users (e.g., AmazonBook, Yelp2018), models with higher $b/a$ values perform better. We hypothesize that in these settings, increasing $b$ reduces reliance on weak cross-item correlations while strengthening self-reconstruction signals.

---

### Official Review · Reviewer_XT64 · 2025-11-01

**Soundness:** 3
**Presentation:** 3
**Contribution:** 3
**Rating:** 6
**Confidence:** 3

**Summary:**

The paper addresses a specific limitation in a previously published model (EDLAE). The main contribution is the extension of the model's analytical solution to a broader set of conditions ($b>0$), which was not covered in the original work. This extension required a derivation and the development of a computationally feasible algorithm. The empirical results, showing improved performance, suggest that this extension is not merely a theoretical exercise but has practical benefits. The findings are of potential interest to researchers working on computationally efficient and effective models.

The paper's claims are supported by the presented derivations and experiments. The derivation of a closed-form solution for the EDLAE objective with $b>0$ is followed step-by-step, and the algorithm to reduce its complexity is well-explained. The experimental setup is described with sufficient detail to understand the procedure: the authors use publicly available datasets, specify the evaluation metrics (Recall@20, NDCG@20), and list the baseline models they compare against. This provides a basis for evaluating the empirical claims.

The paper is structured logically and is generally well-written. It begins with a clear problem statement, moves through the proposed mathematical solution and algorithmic details, and concludes with empirical results. The authors provide context by citing and briefly describing related works in linear autoencoders. However, the clarity could be slightly improved in a few areas. The presentation could be improved by adding more explicit motivation for certain design choices, such as the preference for a closed-form solution. The introduction of the DEQL framework could also be clarified by first showing how the underlying property of decoupling arises from the original objective function's structure.

**Strengths:**

The paper successfully generalizes the closed-form solution of the EDLAE objective from the special case of $b=0$ to the more general case of $b>0$, which increases the model’s flexibility.

The authors developed a practical algorithm for their proposed solution. They designed a method that reduces the computational complexity from $O(n^4)$ to $O(n^3)$, which makes their approach applicable to the datasets used in their experiments.

The claims are supported by experiments on multiple public datasets. The results, as shown in Tables 1 and 2, consistently demonstrate performance gains for the proposed method over the primary baseline (EDLAE) and show its competitiveness against other listed models.

The work provides further evidence that properly formulated linear models can be highly effective for recommendation tasks, sometimes outperforming more complex deep learning models, particularly on sparse data. This is a relevant finding for the field.

**Weaknesses:**

Limited Motivation for the Chosen Approach: The paper does not explicitly justify why a closed-form solution is preferable to a standard gradient-based optimization for this problem. While experts in the subfield might understand the rationale, a broader audience would benefit from a discussion of the trade-offs (e.g., computational cost, guarantee of finding a global optimum, reproducibility).

Lack of Analysis of the Key Hyperparameter: While the authors state in Section 5.2 that they perform a grid search over the hyperparameter $b$ to find the optimal values for their experiments, the paper does not present an analysis of these results. The reader is not shown how the model's performance changes as the $b/a$ ratio is varied. An empirical sensitivity analysis would be necessary to understand the practical importance of this new degree of freedom and to provide guidance on how to tune it.

Potentially Unclear Framing of the "Decoupling": The DEQL framework is presented as a novel formulation. However, the decoupling of the loss function is an intrinsic property of the Frobenius norm and the structure of the EDLAE objective itself. The presentation could be improved by first demonstrating this property in the original objective and then introducing DEQL as a generalized term for such decoupled loss functions.

**Questions:**

Can you elaborate on the practical advantages of finding a closed-form solution versus using an iterative method like SGD for this specific problem, particularly concerning training time, final model performance, and ease of tuning?

How sensitive is the model's performance to the choice of the $b/a$ ratio? Could you provide an empirical analysis (e.g., a plot of performance vs. $b/a$ on a validation set for one or two datasets) to illustrate the impact of this new hyperparameter?

The statistics provided under Tables 1 and 2 are helpful. To further support the discussion on how data characteristics like sparsity influence model performance, could you add a density (or sparsity) column to these existing summary tables? This would allow readers to directly correlate the quantitative characteristics of the datasets with the empirical results.

---

> ### Author Response · Authors · 2025-11-21
>
> Thank you for your insightful comments and constructive suggestions. We hope our responses address your concerns, and we are happy to clarify further if needed.
>
> > W1/Q1: The paper does not explicitly justify why a closed-form solution is preferable to a standard gradient-based optimization for this problem. ... Can you elaborate on the practical advantages of finding a closed-form solution versus using an iterative method like SGD for this specific problem, particularly concerning training time, final model performance, and ease of tuning?
>
> The closed-form solution guarantees the global optimum of the training objective, but it does not necessarily yield the best test performance, as it may overfit the training data. In contrast, gradient-based optimization can rely on *early stopping* to mitigate overfitting, in which case the resulting model does \textit{not} minimize the training objective.
>
> However, the closed-form solution has a distinct advantage in hyperparameter tuning. When hyperparameters are fixed, gradient-based training typically produces non-deterministic models due to the reliance on early stopping, which introduces randomness in addition to the randomness from data splitting. This compounded uncertainty makes it harder to assess whether a particular hyperparameter setting truly leads to good generalization.
>
> In contrast, the closed-form solution is fully deterministic for any fixed hyperparameters, so the only source of randomness stems from the data itself. This substantially reduces uncertainty in model evaluation and makes hyperparameter tuning more reliable.
>
> > W2/Q2: How sensitive is the model's performance to the choice of the ratio $b/a$? Could you provide an empirical analysis (e.g., a plot of performance vs. $b/a$ on a validation set for one or two datasets) to illustrate the impact of this new hyperparameter?
>
> Thank you for raising this important concern.
>
> In **Section 5.4**, we have added experiments analyzing how the ratio $b/a$ influences model performance. By varying $b/a$ in $[0, 2.0]$, we observe that on certain datasets the “sweet spot’’ occurs at $b > 0$ rather than $b = 0$. Surprisingly, in some cases it even occurs at $b/a > 1$ (i.e., $b > a$). Although such settings are not meaningful in EDLAE (as they emphasize non-dropped entries more than dropped ones), they still yield better performance on some datasets. This suggests that emphasizing dropped entries does not universally improve performance and may sometimes hurt it.
>
> Moreover, our analysis suggests that the user-item cardinality ratio (\#items/\#users) can help guide the selection of $b$. For datasets with many items but relatively few users (e.g., AmazonBook, Yelp2018), models with higher $b/a$ values perform better. We hypothesize that in these settings, increasing $b$ reduces reliance on weak cross-item correlations while strengthening self-reconstruction signals.
>
> > W3: The decoupling of the loss function is an intrinsic property of the Frobenius norm and the structure of the EDLAE objective itself. The presentation could be improved by first demonstrating this property in the original objective and then introducing DEQL as a generalized term for such decoupled loss functions.
>
> Thank you for pointing out this clarity issue. We have clarified in **Line 146** that ``decoupling'' arises from an intrinsic property of the squared Frobenius norm.
>
> > Q4: Could you add a density (or sparsity) column to these existing summary tables?
>
> We have added the density for each dataset in **Tables 1 and 2**.

---

> > ### Comment · Reviewer_XT64 · 2025-11-24
> >
> > I thank the authors for their detailed response and for conducting the additional experiments. The authors have successfully addressed my specific concerns:
> > * The experiments in Section 5.4 provide valuable intuition. The finding that the regime $b>a$ can be beneficial for sparse datasets is an interesting insight that justifies the generalized formulation.
> > * The clarification regarding the benefits of a deterministic closed-form solution over iterative SGD (reproducibility, cleaner hyperparameter tuning) is convincing.
> >
> > Regarding the general discussion, I respectfully disagree with the criticism made by Reviewer vBqm concerning the practical relevance of this work. While deep learning approaches are prevalent, Linear Autoencoders (like EASE/EDLAE) remain highly competitive baselines that often outperform complex architectures. Consequently, I believe that refining the mathematical foundation and flexibility of these models is a valuable contribution to the community and fits the scope of the conference.
> > Given the rigorous derivation and the empirical improvements, I consider the paper technically sound and believe it meets the standard for acceptance.

---

> > > ### Author Response · Authors · 2025-11-25
> > >
> > > We sincerely thank Reviewer XT64 for the encouraging and thoughtful feedback. We appreciate your recognition of our additional experiments and the value of the new insights in Section 5.4, as well as your affirmation of the paper’s technical soundness and relevance. We are pleased that your concerns have been fully addressed. Thank you again for your constructive and supportive review.

---

### Official Review · Reviewer_vBqm · 2025-11-01

**Soundness:** 2
**Presentation:** 3
**Contribution:** 2
**Rating:** 4
**Confidence:** 5

**Summary:**

This paper generalizes the Emphasized Denoising Linear Autoencoder (EDLAE) by introducing Decoupled Expected Quadratic Loss (DEQL), which reformulates EDLAE under a statistical expectation perspective and extends the solution space to the full hyperparameter range $b\geq 0$. Unlike the original EDLAE, which only derived a closed-form solution for $b=0$, the authors provide theoretical derivations for $b>0$ and propose an efficient algorithm—based on Miller’s matrix inverse theorem—to reduce computational complexity from $O(n^4)$ to $O(n^3)$ . Experiments on standard recommendation datasets show modest improvements over EDLAE and other linear baselines.

**Strengths:**

1. Clear theoretical generalization of EDLAE. Reformulating EDLAE as an expected loss (DEQL) provides a cleaner statistical foundation and yields closed-form solutions for b>0, which were missing in prior work.

2. Efficient algorithmic adaptation. The use of Miller’s theorem reduces computation from $O(n^4)$ to $O(n^3)$, making the extended solution computable in practice.

3. Empirical improvements. Experimental results show that DEQL with $b>0$ offers consistent but moderate accuracy gains over EDLAE.

**Weaknesses:**

1. Limited novelty in the ICLR context. The main contribution is an extension of an existing model (EDLAE) from $b=0$ to $b>0$. While theoretically meaningful, this is somewhat incremental—more aligned with recommender system venues (KDD, SIGIR, RecSys) than a general ML conference like ICLR.

2. Utilization of existing mathematical tools. The core efficiency improvement relies on Miller’s matrix inverse theorem rather than a new theoretical contribution. The innovation lies in applying this theorem to EDLAE’s structure, not in developing a new algorithmic principle.

3. Scalability remains unsolved. Even after reducing the complexity to $O(n^3)$, the proposed method still requires computing and storing an $N\times N$ item-item co-occurrence dense matrix $R^\top R$, similar to its peers like EASE or SLIM. For large item catalogs, the method remains computationally and memory prohibitive. Thus, the method does not overcome the fundamental scalability bottleneck of linear autoencoders.

4. Limited practical relevance of SAE-based recommenders in modern RS community. While the paper extends the family of Sparse/Linear Autoencoder-based models (SAE4Rec), such methods have largely fallen out of favor in industry-scale recommender systems. In practice, deep learning-based recommenders have generally surpassed linear autoencoders in both representation power and real-world deployment. Although proposes selective empirical results of deep-model baselines, the paper does not address whether DEQL can bridge this gap, scale to large industrial settings, or offer practical advantages beyond incremental improvements over existing SAE models.

**Questions:**

Please refer to Weakness. Besides, the reviewer would also like to discuss the following interpretability potential: Linear/Sparse Autoencoders are recently used as interpretable models rather than competitive recommenders. Could DEQL be used as an explanation module? Does the closed-form solution provide interpretable weight structures or concept mappings?

---

> ### Author Response · Authors · 2025-11-21
>
> Thank you for your thoughtful suggestions on enhancing the clarity and significance of our work.
>
> > W1: The main contribution is an extension of an existing model (EDLAE) from $b = 0$ to $b > 0$. While theoretically meaningful, this is somewhat incremental—more aligned with recommender system venues (KDD, SIGIR, RecSys) than a general ML conference like ICLR.
>
> While our work is motivated by recommender systems, its **core contribution is methodological and theoretical**, situating it squarely within the general **machine learning** context rather than being an application-specific study.
>
> - **General Loss**: As shown in **Definition 3.1**, DEQL permits an arbitrary choice of distribution $\boldsymbol{\mathcal{D}}$, with EDLAE representing only a single special case. This generality makes clear that our DEQL analysis is not confined to EDLAE; it provides a principled framework that can naturally extend to a broad range of LAE models beyond EDLAE.
>
> - **Theoretical and Empirical Discovery beyond EDLAE**: While EDLAE considered only the special case $b = 0$, our work provides the *first general closed-form analysis for arbitrary $b > 0$*, revealing surprising and counter-intuitive behaviors. Specifically, we show that **emphasizing dropped entries (larger $a/b$) does not always improve performance**, and in some datasets, the optimal regime even occurs when $b > a$ (**Section 5.4**). These results challenge a long-held assumption in denoising and self-reconstruction models and would not have been observable without our generalized formulation and efficient analytic solution.
>
> - **New Theoretical Insight**: In **Lines 230-234**, DEQL uncovers the non-uniqueness of the original EDLAE solution with $b=0$. The solution set is infinite, with fixed off-diagonal entries and arbitrary diagonal entries, and the original EDLAE solution corresponds to the zero-diagonal choice. This non-uniqueness suggests that relaxing the zero-diagonal constraint can potentially improve performance, which inspired the design of DEQL(L2).
>
> In summary, our contribution goes well beyond a minor extension of EDLAE. We provide a **general statistical learning framework**, **derive new theoretical insights**, and **uncover novel empirical phenomena** that challenge existing intuitions. These are precisely the kinds of contributions that advance the understanding of learning objectives and generalization, aligning strongly with ICLR’s focus on *fundamental ML principles*.
>
> > W2: Utilization of existing mathematical tools. The core efficiency improvement relies on Miller’s matrix inverse theorem rather than a new theoretical contribution. The innovation lies in applying this theorem to EDLAE’s structure, not in developing a new algorithmic principle.
>
> We respectfully disagree with the claim that our work “merely applies existing mathematical tools without new theoretical contribution.”  As shown in Figure 2 in Appendix A, the closed-form solution of DEQL requires $O(n^4)$ time in the worst case, and whether this complexity can be reduced has not been studied before. We **theoretically** show that, at least in the EDLAE special case, there exists an algorithm that computes the closed-form solution with $O(n^3)$ complexity; we **prove** this by constructing such an algorithm based on Miller's matrix inverse theorem. Thus, while we leverage existing mathematical tools, we use them to obtain new theoretical results.

---

> ### Author Response · Authors · 2025-11-21
>
> > W3: Even after reducing the complexity to $O(n^3)$, the proposed method still requires computing and storing an item-item co-occurrence dense matrix $R^TR$, similar to its peers like EASE or SLIM. For large item catalogs, the method remains computationally and memory prohibitive. Thus, the method does not overcome the fundamental scalability bottleneck of linear autoencoders.
>
> We appreciate the reviewer’s concern regarding scalability.
>
> - **On $O(n^3)$ Complexity** : The reported $O(n^3)$ complexity corresponds to implementations using *basic* linear algebra algorithms for matrix multiplication and inversion. As detailed in **Section 4** and **Appendix C**, our refined analysis shows that the complexity can be reduced to $O(n^{2.376})$ by adopting fast matrix multiplication techniques (e.g., Coppersmith–Winograd), and cannot be improved beyond the theoretical lower bound $\Omega(n^2 \log n)$. These bounds are consistent with the best-known theoretical limits for $n \times n$ matrix operations. Hence, our method matches the asymptotic scalability of existing optimal linear algebraic algorithms.
>
> - **On Memory Prohibitivity**: In practice, storing or computing $R^T R$ is not prohibitive. Modern CPU servers equipped with large RAM (e.g., 512 GB or more) can comfortably accommodate typical recommendation matrices. For GPU environments, where memory is smaller (e.g., 80 GB), we can adopt a **blockwise matrix multiplication** strategy:
> \begin{equation*}
> R^T R =
> \begin{bmatrix}
> R\_{11}^T & R\_{21}^T \\\\
> R\_{12}^T & R\_{22}^T
> \end{bmatrix}
> \begin{bmatrix}
> R\_{11} & R\_{12} \\\\
> R_{21} & R_{22}
> \end{bmatrix}
> \end{equation*}
> where each sub-matrix $R_{ij}$ fits into GPU memory. At each step, only two blocks are loaded, multiplied, and the partial results are aggregated in CPU memory. If needed, finer partitioning (e.g., $3 \times 3$, $4 \times 4$) can ensure scalability to even larger datasets. This design is standard in large-scale linear models such as EASE and SLIM and allows our algorithm to scale comparably.
>
> > W4: Although proposes selective empirical results of deep-model baselines, the paper does not address whether DEQL can bridge this gap, scale to large industrial settings, or offer practical advantages beyond incremental improvements over existing SAE models.
>
> We acknowledge that large industrial recommender systems often rely on deep or hybrid architectures with flexible capacity. The objective of DEQL, however, is not to compete directly with massive-scale deep models, but to bridge the theoretical and practical gap between interpretable linear recommenders and more complex architectures. As discussed in **Section 4** and **Appendix C**, DEQL achieves the best-known computational complexity for exact closed-form linear models $O(n^{2.376})$ with fast matrix multiplication) and can leverage existing scalable EASE implementations for distributed or sparse training. Thus, DEQL remains practical for medium- to large-scale deployments and provides a theoretical foundation that deep architectures can build upon.
>
> > Q: Could DEQL be used as an explanation module? Does the closed-form solution provide interpretable weight structures or concept mappings?
>
> We thank the reviewer for raising important points about both practical scalability and the interpretability potential of DEQL.
>
> We greatly appreciate the reviewer’s suggestion regarding interpretability. Indeed, linear and sparse autoencoders have recently been studied as interpretable concept-extraction models rather than solely as recommenders [1,2]. The DEQL formulation is naturally suited for such use because:
> - it produces a **unique closed-form solution** for each set of hyperparameters, ensuring consistency and reproducibility of interpretability analysis;
> - each entry of the weight matrix $W$ directly encodes item-item (or concept-concept) relationships, making the model’s internal structure transparent; and
> - hyperparameters $(a,b)$ modulate the strength and focus of these dependencies, providing an interpretable control over how the model emphasizes co-occurrence versus self-reconstruction.
>
> In this sense, DEQL can indeed function as an **explanation module**, where the closed-form solution provides an explicit and analyzable mapping between observed behaviors (items) and can be used to *learn or simulate the predictions of complex/deep recommendation models*. Finally, we have also **added a dedicated discussion in the Related Work section** connecting Linear Autoencoders (LAE) with recent literature on model explainability, further situating DEQL within this emerging line of research.
>
> **References**:
>
> [1] Martin Spišák et al. On Interpretability of Linear Autoencoders. Recsys, 2024.
>
> [2] Hoagy Cunningham et al. Sparse autoencoders find highly interpretable features in language models. arXiv preprint arXiv:2309.08600.

---

> > ### Comment · Reviewer_vBqm · 2025-11-24
> >
> > I would like to thank the authors for their detailed rebuttal and the effort in addressing the reviews. I also want to clarify that my intention was **not** to be mean or overly negative, especially relative to the other three reviewers who were generally more positive. My initial expectation was to discuss certain aspects of scope, positioning, and practicality that I believe are important for the paper’s presentation.
> >
> > - On W4, I agree that SAE models are still widely used as comparable baselines in industrial recommendation systems due to their structural stability and simplicity of deployment, and that in some specific regimes like cold-start they can even outperform more complex deep models. My intent in raising W4 was not to deny this consensus, but to encourage the authors to explicitly and positively situate their method: to show that they understand where and how DEQL fits into real-world utilizations, and to strengthen the “story” of the paper.
> >
> > - On W1 & Q1, as a researcher of RecSys, I have repeatedly encountered similar questions at general ML venues regarding whether a contribution is “too specific to recommendation,” even when the work contains broadly applicable theoretical analysis. I therefore appreciate the authors’ claim that DEQL should be viewed as a generalized methodological and theoretical contribution rather than a purely RecSys-specific technique. However, if the authors wish to argue that the work situates firmly in a general ML context, I believe their argument should go beyond RecSys-specific language. In particular, I see two possible directions that would strengthen this claim: i) show applicability beyond recommendation by providing at least one additional application domain where DEQL formulation is directly meaningful; or ii) if the authors maintain recommendation as the primary application, they should explicitly argue why this domain is particularly well suited to showcasing DEQL , while still clarifying that the framework is in principle applicable elsewhere.
> >
> > - On W2, after revisiting the paper in light of the rebuttal, I am convinced that the work is not merely an application of an existing theorem.
> >
> > - On W3, the authors provide a detailed complexity analysis and connect their results to fast matrix multiplication bounds. I agree that, from a purely theoretical perspective, it is meaningful to show that the complexity matches the best-known asymptotic bounds for dense $N\times N$ matrices. At the same time, in practical industrial implementations, these fast matrix multiplication algorithms are rarely used due to their implementation complexity. However, I do appreciate the authors’ introduction of ELSA with possible approximate / low-rank strategies for controlling computational cost. I see this as a promising direction for future work.
> >
> > In summary, I thank the authors again for their careful rebuttal. While I still have some open questions about how to best present the scope and the practical scalability story, I believe many of my original concerns have been at least partially addressed, and I encourage the authors to make further responses with my updated comments.

---

> > > ### Author Response · Authors · 2025-11-25
> > >
> > > We greatly appreciate the clarification of intent and the specific guidance on strengthening our paper's positioning and scope.
> > >
> > > We are glad that our responses have addressed your concerns on **W2**  (contribution beyond existing theorems). We also appreciate the positive feedback on ELSA and the low-rank strategies for practical scalability. Regarding **W4**, we thank the reviewer for the clarification.
> > >
> > > Below, we provide a detailed response to the main open question regarding  **W1 \& Q1** (scope) and **W3** (practical scalarbility).
> > >
> > > > **On W1 \& Q1** (scope)
> > >
> > > We follow the two constructive directions suggested by the reviewer, addressing ii) first and then i):
> > >
> > > **1. DEQL as a general framework, with recommendation as a strategic testbed**:
> > >
> > > Our primary motivation is to provide a principled theoretical framework  for LAE-based recommenders -- a class of models that, despite their widespread practical use, remains largely guided by intuition and empirical heuristics. DEQL offers the first expected-loss formulation for LAE-style models, moving beyond the traditional ``matrix-completion'' viewpoint and enabling the introduction of training-evaluation consistent objective function (formalizing random dropout to align with the hold-out evaluation mechanism) and the study of previously underexplored $b>0$ regime.
> > >
> > > **Crucially, DEQL is a general analytical framework**. As illustrated in Figure 2 in Appendix A, the family of distributions $\\{\mathcal{D}^{(i)}\\}\_{i=1}^n$  is flexible and not inherently tied to any specific application. EDLAE  represents only one particular instantiation; the framework in principle  supports other distribution choices and could enable principled design of alternative models for various domains.
> > >
> > > We chose recommender systems as our primary testbed for several reasons: (1) LAE-based recommenders are widely used yet lack theoretical foundations, (2) they suffer from training-evaluation inconsistencies that DEQL directly  addresses, and (3) they provide established benchmarks for empirical  validation. This makes recommendation an ideal domain for demonstrating
> > > DEQL's theoretical and practical contributions *while maintaining  broader applicability*.
> > >
> > > **2. Applicability of DEQL beyond recommendation**:
> > >
> > > Beyond recommender system, DEQL can be potentially used in *Online Linear Regression Learning* [1, 2] or *Distributed Linear Regression Learning* [3, 4]. The formulation of DEQL inherently captures their core feature: each data pair $(X\_i, Y\_i)$ in the loss $\frac{1}{m}\sum\_{i=1}^m\|\|Y\_i - X\_iW\_{\*i}\|\|\_F^2$ can follow a different distribution. In online learning, this arises because samples arrive sequentially and depend on earlier observations; in distributed learning, each party maintains its own data subset with potentially distinct distributions. More broadly, whenever a scenario requires $(X\_i, Y\_i)$ to follow heterogeneous distributions across $i$, the DEQL framework can naturally be adapted.
> > >
> > > **References**:
> > >
> > > [1] Alexander Strehl and Michael Littman. Online linear regression and its application to model-based reinforcement learning. NIPS, 2007.
> > >
> > > [2] Yujing Liu et al. Convergence of online learning algorithm for a mixture of multiple linear regressions. ICML, 2024.
> > >
> > > [3] Edgar Dobriban and Yue Sheng. Distributed linear regression by averaging. The Annals of Statistics, 2021.
> > >
> > > [4] Martin Hellkvist et al. Linear regression with distributed learning: A generalization error perspective. IEEE Transactions on Signal Processing, 2021.

---

> ### Author Response · Authors · 2025-11-25
>
> > **On W3** (practical scalability)
>
> Thank you again for your concern on practical algorithm scalability! We use the CW algorithm in the proof to obtain an $O(n^{2.376})$ complexity, but we would like to clarify that this reflects **theoretical** scalability, not **practical** scalability. In practice, the $O(n^{2.376})$ asymptotic advantage only emerges when $n$ is extremely large (Figure 3.1, [5]), and current industrial scenarios are **unlikely** to reach a regime where an $O(n^{2.376})$ method is practically faster than a standard $O(n^3)$ implementation.
>
> The practical scalability issue arises much earlier: one typically hits **memory prohibitivity** long before the $O(n^{2.376})$ time complexity becomes the bottleneck. As mentioned in our previous response, when an $n\times n$ model fits in CPU RAM but is GPU-memory prohibitive, standard block-wise linear-algebra routines (multiplication, inversion, etc.) can partition the computation into memory-fitting blocks, resolving this issue.
>
> We add one more point here: in even more restrictive memory environments where storing a full-rank $n\times n$ model in RAM is infeasible, a low-rank LAE model can be used instead -- for example, the ELSA (Appendix G.3) model obtained by taking $l < n$. We also provide a **low-rank DEQL** solution in Appendix D.
>
> Finally, for practical implementation, one can simply replace the $O(n^3)$ routine with Strassen’s algorithm ($O(n^{2.81})$), which remains practical to implement -- unlike the CW algorithm. Our use of CW is solely to obtain the **best theoretical complexity bound**; it is not intended for practical deployment.
>
> **Reference**:
>
> [5] Cormen et al. Introduction to Algorithms, 3rd Edition.
>
> &nbsp;
>
> We hope this clarification addresses your concerns, and we thank you again for helping improve our paper!

---

> > ### Comment · Reviewer_vBqm · 2025-11-25
> >
> > Thank you for your response. I have carefully read your latest rebuttal. However, I regret to say that I do not feel that authors have fully addressed some of the core concerns from my previous comments:
> >
> > - On application domains beyond recommender systems:
> > In my previous response, my intention was to ask for examples where DEQL can be directly applied outside RecSys setting. In this round, the authors provided “online linear regression / distributed linear regression” as examples. However, from a categorical perspective, recommender systems constitute a specific application domain, whereas online/distributed linear regression are more appropriately viewed as modeling or learning paradigms. They are not at the same level of abstraction. In fact, one can certainly apply linear regression methods within RecSys scenarios. Thus, presenting “RecSys” and “Online/Distributed Linear Regression” as parallel “application domains” may lead to a confusion of conceptual levels.
> >
> > - On the clarification of W4:
> > My previous response explicitly stated that I do not deny the practical value of SAE/LAE models in certain industrial scenarios. In particular, for specific tasks such as cold-start, linear or sparse models can even outperform deep models. My suggestion was that the authors could explicitly acknowledge and elaborate on this point, so as to more clearly position DEQL. The authors seem to have missed this part and concentrate more on the significance of DEQL from other perspectives.
> >
> > Overall, I remain positive about the theoretical aspects of your work. However, in terms of the paper’s scope and positioning, I still feel that some of my concerns have not been fully resolved, so I would like to keep my original score at present.

---

> ### Author Response · Authors · 2025-11-27
>
> We sincerely thank the reviewer for the additional clarifications and for the continued engagement with our work. We appreciate the constructive nature of the feedback and agree that several points in our previous responses should have been articulated more clearly. We address both remaining concerns below and will update the manuscript accordingly.
>
> > **On Application Domains Beyond Recommender Systems**
>
> To address your point regarding \emph{application domains} (not \emph{learning paradigms} (e.g., online or distributed regression), here, we provide several non-RecSys application domains, where
> LAE-style models and the DEQL framework can be potentially applied:
>
> - **Survey \& psychometric modeling; genomics \& biomedical panels**: In survey research, user $\times$ item response matrices exhibit substantial missingness and heterogeneous exposure, a setting long modeled using linear or matrix-completion methods [1,2]. Similarly, genomics, metabolomics, and biomedical panels routinely rely on linear or low-rank imputation methods for patient~$\times$~gene and panel-level data, including mass-spectrometry–based metabolomics [3,4].
>
> - **Distributed Sensing \& Sensor Network**:  In distributed sensing applications, sensor~$\times$~time matrices frequently contain missing measurements due to intermittent connectivity, power constraints, or sensor failures. Linear reconstruction and imputation methods continue to be widely used in these resource-constrained environments, where computational simplicity and interpretability are critical [5].
>
> - **LLM Interpretability**:  Many interpretability methods use linear mappings of the form $Y \approx WX$, including concept-extraction SAEs, probing classifiers, and certain linearized attention approximations. These directly match DEQL’s weighted linear reconstruction structure (See Appendix E).
>
> Finally, we note a conceptual connection to LLM preference learning: while methods such as DPO differ substantially in mathematical formulation, both LLM content generation and recommendation involve selecting or ranking items from large discrete spaces based on preference signals [6,7].
>
> Although direct application would require substantial methodological development beyond DEQL's current scope, this connection suggests an interesting direction for future research.
>
> **References**:
>
> [1] Rahul Mazumder et al. Spectral Regularization Algorithms for Learning Large Incomplete Matrices. JMLR, 2010.
>
> [2] Jinsung Yoon et al. GAIN: Missing Data Imputation using Generative Adversarial Nets. ICML, 2018.
>
> [3] Daniel J. Stekhoven and Peter Bühlmann. MissForest--non-parametric missing value imputation for mixed-type data.
> Bioinformatics, 2012.
>
> [4] Renjie Wei et al. Missing value imputation approach for mass spectrometry–based metabolomics data. Scientific Reports, 2018.
>
> [5] Rivera-Muñoz et al. Deep matrix factorization models for estimation of missing data in a low-cost sensor network to measure air quality. Ecological Informatics, 2022.
>
> [6] R Rafailov et al. Direct preference optimization: Your language model is secretly a reward model. arXiv:2305.18290.
>
> [7] Steffen Rendle et al. BPR: Bayesian Personalized Ranking from Implicit Feedback. UAI, 2009.

---

> ### Author Response · Authors · 2025-11-27
>
> > **Addressing Concerns of W4**
>
> We thank the reviewer for the helpful clarification. We now explicitly acknowledge the practical value of linear and sparse autoencoder models (e.g., EASE, EDLAE, SAE) in real-world recommender systems. These models remain widely used because of their simplicity, robustness, interpretability, and strong performance in cold-start or low-information regimes (e.g., users with short histories) [1,2]. As the reviewer noted, in such settings linear or sparse models can even outperform deep architectures, and we will revise the manuscript to make this positioning clear.
>
> Moreover, many small-to-medium-scale e-commerce platforms possess abundant interaction logs but limited or no side features -- a scenario that characterizes a substantial portion of industrial recommendation deployments. In these environments, LAE-style models and matrix factorization continue to be effective and operationally attractive.
>
> Finally, DEQL also serves as an interpretability and diagnostic tool for broader recommendation pipelines, including deep models. The DEQL item–item affinity matrix $W$ captures co-occurrence and influence patterns that support practical use cases such as “frequently bought together” recommendations, promotional bundling, and cross-selling workflows[4]. These interpretable relationships form a major reason why LAE-style models remain prominent in production systems.
>
> We sincerely appreciate the reviewer’s feedback; The clarifications you raised have substantially improved our understanding of how to present the paper more effectively. We look forward to updating the manuscript once we receive your confirmation that the above responses resolve the outstanding concerns.
>
> **References**:
>
> [1] Monteil et al. MARec: Metadata Alignment for cold-start Recommendation. arXiv:2404.13298.
>
> [2] Maksims Volkovs et al. DropoutNet: Addressing Cold Start in Recommender Systems. NeurlIPS, 2017.
>
> [3] Maurizio Ferrari Dacrema et al. Are We Really Making Much Progress? A Worrying Analysis of Recent Neural Recommendation Approaches. RecSys, 2019.
>
> [4] Vojtěch Vančura et al. Scalable Linear Shallow Autoencoder for Collaborative Filtering. Recsys, 2022.

---

> > ### Comment · Reviewer_vBqm · 2025-11-27
> >
> > Thank you for your response. While I still hold reservations regarding the current manuscript in our discussions, I have raised my score to 6 to acknowledge the authors' efforts during the rebuttal. I wish the authors upload their revised manuscript within the rebuttal period :)

---

### Author Response · Authors · 2025-11-21
**Summary of Changes**

We sincerely thank all reviewers for their insightful, constructive, and encouraging feedback. Your comments have greatly helped us clarify our contributions and strengthen the overall quality of the paper. We have carefully revised the manuscript to address each point in detail and incorporated your suggestions throughout. The main changes are summarized below:

- In **Line 139**, we clarified the meaning ``decoupling'', as suggested by Reviewer **XT64**.
- In **Theorem 3.3**, we generalize the condition ensuring the positive-definiteness of $G^{(i)}$ from $a \ge b > 0$ to $a \ge 0, b > 0$. The corresponding proof in **Appendix B** has been updated. This extension means that the closed-form solution of DEQL remains valid even when $b > a$, which goes beyond the hyperparameter restriction $a \ge b > 0$ in EDLAE.
- To address the $O(n^3)$ complexity concern raised by Reviewers **9r8b** and **vBqm**, we provide a refined complexity analysis in **Section 4** and **Appendix C**. The analysis shows that the complexity can be reduced to $O(n^{2.376})$, but cannot be improved beyond $\Omega (n^2\log n)$. We further note that recent advances in scalable EASE implementations, such as ELSA, can be applied to our framework through techniques such as low-rank factorization, approximate matrix inversion, and parallelized block-wise computation.
- To answer how the hyperparameter $b$ affects model performance -- raised by Reviewers **9r8b**, **bLSH** and **XT64** -- we empirically conduct a sensitive analysis on $b$ in **Section 5.4**. By varying $b/a$ in $[0, 2.0]$, we observe that on certain datasets the “sweet spot’’ occurs at $b > 0$ rather than $b = 0$. Surprisingly, in some cases it even occurs at $b/a > 1$ (i.e., $b > a$). Although such settings are not meaningful in EDLAE (as they emphasize non-dropped entries more than dropped ones), they still yield better performance on some datasets.  This suggests that emphasizing dropped entries does not universally improve performance and may sometimes hurt it. We also include an extended discussion on potential underlying factors that could explain these counterintuitive results.
- In **Tables 1 and 2**, we added dataset densities (as suggested by Reviewer **XT64**) and included the recent LAE model ELSA (Vancura et al., 2022) for comparison (as suggested by Reviewer **bLSH**).
- In **Appendix A**, we added a figure illustrating the DEQL framework, following the suggestion from **bLSH**.
- In **Appendix E**, we added discussion on the relationship between linear autoencoder and model explainability.
- In **Appendix F.1**, we added the statistical significance tests requested by Reviewer **9r8b**. Since for any fixed $b$, the closed-form solution is deterministic (and unique), the only source of randomness is dataset splitting. We evaluate each model on five random splits of each dataset, compute standard deviations of the test errors, and perform paired t-tests. The results indicate that DEQL(L2)’s improvements are statistically significant.
- In **Appendix F.2**, we added a visualization of the diagonal elements of the DEQL(L2) model, as requested by Reviewer **9r8b**. The results show that most diagonal values are close to 0, indicating that relaxing the zero-diagonal constraint to allow small values can improve performance.
- In **Appendix G.3**, we clarified the scalability of LAE models.

Please let us know if any of your concerns remain unaddressed. We will update the revision accordingly.

&nbsp;

=====================================UPDATES=========================================

Thanks to the reviewers for their thoughtful comments and engagement. We have uploaded a new revision that includes additional discussions and clarifications. Building upon the previous version, we have made the following updates:
- In **Lines 274-276**, we clarified the theoretical contribution of our Fast Algorithm, addressing the concern raised by Reviewer **vBqm**.
- In **Lines 380-382**, we clarified that all LAE baselines in our experiments are already equipped with $L2$ regularization, addressing the concern raised by Reviewer **9r8b**.
- We extended the discussion in **Appendix G.3** to cover the practical scalability of DEQL, addressing the concern from Reviewer **vBqm**.
- We added **Appendix G.4**, which discusses the advantages of closed-form solutions to further highlight the significance of DEQL, as suggested by Reviewer **XT64**.
- We added **Appendix G.5** comparing LAEs and deep models, demonstrating that LAEs outperform deep models on sparse datasets and can also serve as tools for interpretable AI, as suggested by Reviewer **vBqm**.
- We added **Appendix G.6** discussing the broader applicability of LAEs and DEQL and outlining potential use cases beyond the scope of recommender systems, as suggested by Reviewer **vBqm**.

We acknowledge that reviewers can no longer respond, and we hope these updates adequately address their concerns.

---

### Meta-Review · Area_Chair_Bci5 · 2025-12-17

**Summary:**

This paper has seen heated discussions focused primarily on the scope and positioning of the contribution, the practical scalability of the proposed closed-form solution, and the empirical significance of the reported gains.

While some reviewers initially viewed the work as an incremental extension of EDLAE or questioned its fit within a general ML venue, the rebuttal clarified that DEQL is a principled expected-loss framework that generalizes beyond a single recommender formulation and yields new theoretical and empirical insights.

 Additional experiments, sensitivity analyses, statistical tests, and comparisons to stronger baselines substantially strengthened the empirical case. Overall, the reviewers seem to have converged on recognizing the technical soundness, clarity, and value of the contribution.

**Reviewer Concerns:**

I believe most substantive concerns were addressed during rebuttal.

Requests such as HP sensitivity analysis, statistical significance testing, clearer explanation of the “decoupling” concept, stronger empirical baselines, and improved presentation etc., were all resolved. Concerns about scalability and practical relevance were mitigated by refined complexity analysis, and discussion of blockwise and low-rank strategies, within regimes where linear autoencoders remain competitive (e.g., sparse and cold-start settings).

One outstanding concern regarding broader application framing beyond recommender systems remains partially subjective, but it does not detract from the paper’s core technical contributions.

**Reviewer Scores:**

AC believes most reviewers would likely have increased their scores after the rebuttal, moving from borderline accept to a clearer accept. The most critical reviewer would likely have shifted from a marginally negative or borderline score to a marginal accept, as indicated in their own discussion thread, reflecting partial resolution of concerns and acknowledgment of the authors’ substantial effort.

---

### Decision · Program_Chairs · 2026-01-26

Accept (Poster)